

# Technical note: An experimental setup to measure latent and sensible heat fluxes from (artificial) plant leaves

Stanislaus J. Schymanski[1], Daniel Breitenstein[1], and Dani Or[1]

[1]Department of Environmental Sciences, ETH Zurich, 8092 Zurich, Switzerland

*Correspondence to:* Stan Schymanski (stan.schymanski@env.ethz.ch)

**Abstract.** Leaf transpiration and energy exchange are coupled processes that operate at small scales yet exert a significant influence on terrestrial hydrological cycle and climate. Surprisingly, experimental capabilities required for quantifying the energy-transpiration coupling at the leaf scale are lacking, challenging our ability to test basic questions of importance for resolving large scale processes. The present study describes an experimental setup for the simultaneous observation of transpiration rates and all leaf energy balance components under controlled conditions, using an insulated closed-loop miniature wind tunnel and artificial leaves with pre-defined and constant diffusive conductance for water vapour. A range of tests documents the above capabilities of the experimental setup and points to potential improvements. The tests reveal a conceptual flaw in the assumption that leaf temperature can be characterised by a single value, suggesting that even for thin, planar leaves, a temperature gradient between the irradiated and shaded or transpiring and non-transpiring leaf side can lead to bias when using observed leaf temperatures and fluxes to deduce effective conductances to sensible heat or water vapour transfer. However, comparison of experimental results with an explicit leaf energy balance model revealed only minor effect on simulated leaf energy exchange rates by the neglect of cross-sectional leaf temperature gradients, lending experimental support to our current understanding of leaf gas and energy exchange processes.

## 1   Introduction

Most of precipitation falling on land returns to the atmosphere by the process of transpiration, i.e. passing through the plant vascular system, undergoing phase change in leaves and diffusing through stomata. Plant transpiration rates and $CO_2$ uptake are controlled by stomata and by the leaf energy balance, i.e. the partitioning of the absorbed solar irradiance into radiative, sensible and latent heat fluxes. Present understanding of leaf gas and energy exchange is based on controlled experiments with real and artificial leaves, where the individual components of the energy balance and their sensitivities to environmental forcing were assessed separately. The state of the art measurements of leaf transpiration rates employ a mass balance of an open controlled volume at steady state, i.e. by the difference of the products of air flow rate and humidity between the incoming air and the outgoing air from a control volume containing a transpiring leaf (Field et al., 1982). The transfer of heat between a leaf and the surrounding air is less commonly measured. Studies exist where this heat flux was estimated from cooling curves after a sudden reduction in absorbed radiation (Kumar and Barthakur, 1971), but in order to test our understanding of the leaf energy balance, we need a way to monitor leaf heat and gas exchange simultaneously under controlled steady state conditions.





Leaf gas exchange and hence the leaf energy balance underlies strong biological control by stomata. Only few studies exist where leaf gas exchange and stomatal apertures were simultaneously observed (e.g. Kappen et al., 1987; Kaiser and Kappen, 1997), but these observations were not used to study physical processes, probably due to strong dynamics and uncertainty related to deduction of stomatal conductance from observed apertures. Therefore, many studies employed leaf replica to gain better understanding of individual processes related to the leaf energy balance and gas exchange separately from biological control. For example, externally or internally heated plates were employed to estimate sensible heat transfer coefficients as a function of plate size/shape, wind speed and turbulence (Wigley and Clark, 1974; Thom, 1968; Parkhurst et al., 1968; Grace et al., 1980). Others have used wetted leaf replica and weighing or electrochemical methods using leaf-shaped electrodes to obtain mass transfer coefficients (Schuepp, 1972). In the latter method, dimensional analysis was used to transfer results obtained from a liquid medium to real leaves surrounded by air.

A range of studies employed micro-perforated foils or plates to study the effect of pore size and density on transpiration under steady state conditions (e.g. Brown and Escombe, 1900; Sierp and Seybold, 1929; Ting and Loomis, 1963; Cannon et al., 1979). In most of these experiments, the perforated surface was mounted on a water reservoir and transpiration rate was measured by weighing the water reservoir. So far, studies using artificial leaves with dimensions and pore sizes similar to real leaves have not been published. Morrison Jr and Barfield (1981) presented a thin artificial leaf design of similar shape and size as a tobacco leaf, consisting of teflon membrane disks sandwiching a filter paper and an external water supply consisting of cotton wicks, with a total leaf thickness of 0.42 mm. This could easily be combined with some of the above-mentioned perforated foils in order to obtain a more realistic physical model of a real leaf, but surprisingly, we have not found any such experiments in the literature.

Even more suprisingly, the simultaneous measurements of radiative, latent and sensible heat exchange of transpiring leaves or leaf replica have not been presented in the literature. This suggests that leaf energy balance closure has never been used to assess uncertainty in the observations in a similar way as is commonly done for eddy covariance measurements at the canopy scale (e.g. Wohlfahrt and Widmoser, 2013). In contrast to the canopy scale, leaf-scale processes lend themselves to investigation through controlled experiments, theoretically permitting rigorous testing of our understanding of leaf energy partitioning, which is at the basis of canopy-scale processes.

To improve our experimental and observational capabilities at the leaf scale, the goal of the present study was to design an experimental setup that permits the direct measurement of all the leaf energy components simultaneously (sensible, latent and radiative exchange) while controlling boundary conditions (air temperature, humidity, wind speed, irradiance).

## 2 Materials and Methods

To separate the physical aspects of leaf energy and gas exchange from biological control, we used artificial leaves with laser-perforated surfaces representing fixed stomatal apertures and embedded thermocouples to obtain the best possible measurements of leaf temperature near the evaporating sites (Fig. 1). We further devised a specialised thermally insulated leaf wind tunnel to control atmospheric conditions including air temperature, humidity, irradiance and wind speed and allowing mea-





surement all leaf energy balance components independently, including net radiation, as well as latent and sensible heat flux

(Fig. 2). The leaf wind tunnel and the artificial leaves are described in detail below. Details of technical equipment used in this study are listed in Table A2. All variables used in this paper, their descriptions, units and standard values are given in Table A3. All data, equations and model code necessary to reproduce the results presented here can be accessed online at: https://github.com/schymans/Schymanski_experimental_2016.git

## 2.1 Artificial leaves

Different artificial leaves were constructed, all consisting of a capillary filter paper glued onto aluminium tape, with a water supply tube and a thermocouple sandwiched between the filter paper and the aluminium tape. The water supply tube was flattened at the end and tightly glued to the aluminium tape and filter paper using Araldite epoxy resin (Fig. 1), to prevent intrusion of air along the edges. For some leaves, we used Whatman No. 41 filter paper (0.2 mm thick) and embedded 0.25 mm thick copper-constantan thermocouples (TG-T-30-SLE, Tab. A2), whereas for others, we used 0.1 mm thick Durapore membrane

filter (Type 0.45$\mu$m HV[1]) and 0.08 mm thick thermocouple wire (TG-T-40-SLE). The Durapore membrane filters appeared more homogeneous and tear resistant than the Whatman filter papers, whereas the thinner thermocouple wires produced smaller bumps on the leaf surface. The water supply was connected to a liquid flow meter (SLI-0430, Tab. A2) and a water supply with a free water surface placed 1-3 cm below the position of the leaf. It had to be lower than the leaf to ensure that the liquid flow did not exceed the transpiration rate (e.g. droplets forming due to positive head between reservoir and leaf) and as high as

possible to avoid cavitation and air intrusion along the flow path. Stomatal resistance was introduced by covering the surface of the capillary filter paper with a laser-perforated aluminium foil, attached to the leaf using thin strips of double-sided sticky tape lining the outside of the rectangular leaves. The laser-perforated foils were untreated aluminium of 25 $\mu$m thickness. Laser perforations of different sizes and densities produced different effective leaf conductances. Laser perforations were performed by Ralph Beglinger (Lasergraph AG, Würenlingen, Switzerland). The geometry of the laser perforations used for each leaf was

measured using a confocal laser scanning microscope (CLSM VK-X200, Keyence, Osaka, Japan) and the specific diffusive conductances for the perforated surfaces were estimated based on derivations presented by Lehmann and Or (2015), neglecting any internal resistance (termed "end correction" by Lehmann and Or (2015)), as we assume that the wet filter paper has direct contact with the perforated foil. The relevant equations are described in Appendix A.

## 2.2 Thermal mapping of artificial leaves

To evaluate the spatial temperature distribution of the artificial leaf surface and to assess in how far the temperature recorded by the embedded thermocouple may be seen as representative of average leaf temperature, we recorded infrared images of the leaf surface temperature. For this purpose, artificial leaves were placed in a conventional wind tunnel above a heat plate linked to a water bath, thereby providing constant background temperature, and infrared images of the leaf surfaces were obtained using a cryogenically cooled infrared (IR) camera (FLIR SC6000, Tab. A2) at different wind speeds. For these experiments, we

did not use laser-perforated foils, but exposed the wet (or dry) filter paper directly to the IR camera, when the leaf was placed

---

[1]Ref. HVLP04700, www.merckmillipore.com



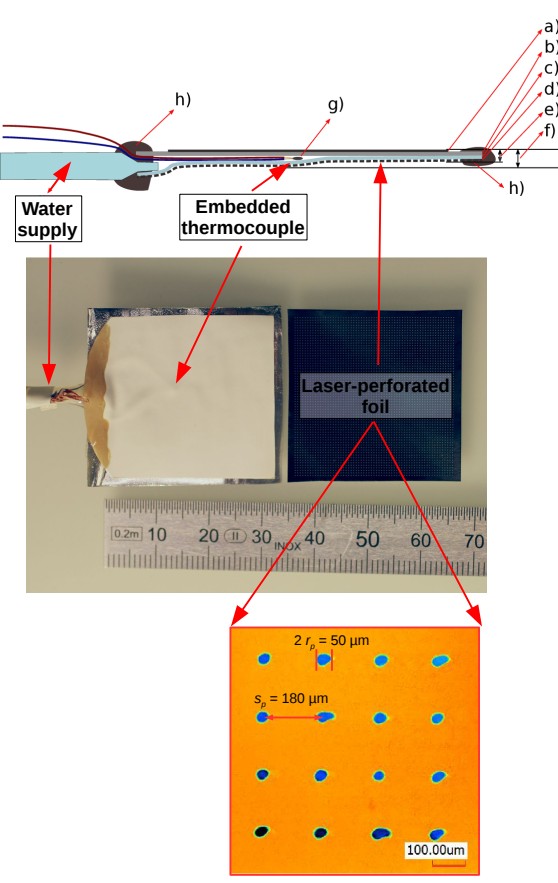

**Figure 1.** Artificial leaf. Top: cross-section of artifical leaf; center: leaf image before full assembly; bottom: topography of a laser-perforated foil obtained using confocal laser scanning microscopy (CLSM). a) black aluminium tape (0.05 mm thick); b) aluminium tape (0.08 mm); c) absorbent filter paper (0.1-0.2 mm); d) laser-perforated foil (0.025 mm); e) min. leaf thickness: 0.3-0.4 mm; f) max. leaf thickness: 0.35-0.65 mm; g) thermocouple; h) glue; i) water supply tube (from flow meter). Numbers in CLSM image indicate typical pore diameter ($2r_p$) and pore spacing ($s_p$) for foil with 7 perforations per mm$^2$.





with the evaporating side upwards. We also reversed the leaf to detect any differences in surface temperature between the two leaf sides.

## 2.3 Leaf wind tunnel

For measurements of the water vapour and energy exchange of artificial leaves under fully controlled conditions, we designed
a thermally insulated closed loop wind tunnel with a transparent leaf chamber, allowing control of gas and energy exchange with the surroundings (Fig. 2), as described below.

The main body of the wind tunnel was built of extruded polystyrene foam slabs (Sagex XPS-EN13164-T3-CS, Sager AG, Dürrenäsch, Switzerland[2]) with a low heat capacity (1400 J kg$^{-1}$ K$^{-1}$ at a density of 30 kg m$^{-3}$) and low thermal conductivity (0.035 W m$^{-1}$ K$^{-1}$). The geometry and dimensions of the wind tunnel are given in Fig. A4. It is a closed loop tunnel with
rectangular inner cross-section, which varied gradually between $5 \times 3$ cm in the leaf chamber and $5 \times 5$ cm on the opposite site, where a fan occupies the entire cross-section (Fig. 2).

The frame of the transparent leaf chamber was produced by a 3D printer, using transparent acrylic rasin. The walls consist of three layers of 1 mm thickness, separated by 1 mm thick air gaps. On the inner walls of the chamber, a 1 cm thick layer of polystyrol foam was added for improved thermal insulation. At the top and bottom, the chamber was sealed with two
layers of transparent PVC coated polypropylene (Propafilm®-C, ICI Americas Inc., Wilmington, DE, USA). The two layers of polypropylene foil were intended to permit the transmission of shortwave and longwave radiation while minimising conductive heat transfer.

The net radiation of the leaf was measured using Peltier-based heat flux sensors of 1 cm by 1 cm size (gSKIN, Tab. A2), which were painted black and calibrated against a net radiometer (NR Lite2, Tab. A2) using a tungsten light source. The sensor
response to irradiance varying between 0 and 700 W m$^{-2}$ was linear ($R^2 > 0.99$) and the sensitivity ranged between 0.001 and 0.0013 mV per W m$^{-2}$ net radiation. Three of these heat flux sensors were mounted on retractable wires such that they could be periodically positioned above, beside and below the artificial leaf for radiation measurement, while being kept out of the chamber during equilibration. Their positions during a measurement were 1 cm above the leaf, 1 cm below the leaf and one was positioned at the same height as the leaf, but 0.5 cm downwind from the leaf (Figs. 3 and 4).

Temperature measurements were performed using T type thermocouples (Tab. A2), which were placed (a) in the air stream upstream and downstream of the leaf chamber, (b) lightly inserted into the wind tunnel wall on the inside and the outside of the chamber, and (c) in the duct through which air was supplied to the wind tunnel by a humidifier. The air humidifier was a custom assembly by Cellkraft (Tab. A2) and provided an adjustable flow rate of up to 10 l/min, with adjustable air temperature and dew point. Air temperature was controlled by an external chiller (MRC300DH2-HT-DV, Laird Technologies, Cleveland
OH, USA), supplying the humidifier with cooling liquid (water) between 4$^o$C and 40$^o$C.

Constant wind speed was generated using an axial fan of 5 cm by 5 cm diameter (MULTICOMP - MC35357, Tab. A2), which produced wind speeds of up to 5.4 m s$^{-1}$ at a power consumption of less than 1.4 W in our wind tunnel, compared to a sensible heat exchange of up to 0.6 W by our 3 by 3 cm artificial leaves. A stack of 3 cm long plastic straws (each with

---

[2]http://www.sager.ch




diameter of 7 mm) in the flow path acted as straighteners to reduce spiralling of the air flow caused by the rotating fan. The fan was placed inside the chamber, enabling direct control over the amount of heat injected by the fan into the wind tunnel, deduced from its rate of electrical power consumption. Power consumption by the fan was kept constant using a programmed power controller (NI USB-6008, Tab. A2), while wind speed was varied by adjusting the position of a flap in the flow path (Fig. 2) and monitored by a miniature thermal wind speed sensor (FS5 Flowmodule, Tab. A2), which was calibrated in the wind tunnel against a high accuracy air flow sensor (EE75, Tab. A2). The calibration produced a non-linear relationship between recorded sensor voltage and wind speed, ranging from 1350–1425 mV at 1.2 m s$^{-1}$ wind speed to 1537–1630 mV at 4.4 m s$^{-1}$ wind speed for different sensors. For each sensor, we fitted an exponential relationship between wind speed and voltage with an R$^2$ > 0.99, which had the tendency to over-estimate wind speed at values above 4.2 m s$^{-1}$ and below 1.2 m s$^{-1}$. The wind speed sensors were only turned on briefly after each recording of chamber steady state conditions to avoid contamination of the air temperature signal by the sensors' heat production.

All devices were connected to data loggers (CR 1000, Tab. A2), logged every second and plotted on computer screens. Steady states were identified visually be examining the graphs and data points were generated by averaging the values of each sensor over 10 seconds at steady state.

### 2.4 Calculation of sensible heat flux

The exchange of sensible heat between the artificial leaf and the air was calculated based on the energy balance of the entire wind tunnel, by difference between the heat contained in the incoming and in the outgoing air (Fig. 5) and subtracting the heat added by the fan (see Appendix, B for details on the thermodynamic calculations). The thermal insulation of the wind tunnel minimised uncontrolled heat exchange with the surroundings. To estimate the rate of conductive heat exchange per temperature difference between the air inside and outside the chamber, we considered the entire air-wall interfacial area at the inner side of the tunnel totalling 868 cm$^2$ and a minimum wall thickness of 5 cm (Fig. A4). This would result in a conductive heat transfer per Kelvin chamber-lab air temperature difference of roughly: 0.035 W m$^{-1}$ K$^{-1}$/0.05 m ×0.0868 m$^2$ = 0.061 W K$^{-1}$. Considering that a $3 \times 3$ cm large leaf exchanges 0.09 W heat with the chamber air per 100 W m$^{-2}$ sensible heat flux, 1 K temperature difference between the chamber air and the lab air would roughly add a bias of 70 W m$^{-2}$ in our estimation of sensible heat flux. To reduce the impact of this potential bias, we regulated the air temperature within the windtunnel to track the external air temperature in the lab to within ±0.1 K.

### 2.5 Leaf gas and energy exchange model

The transpiration rates and energy balance measurements were compared with model simulations based on a steady state solution of the leaf energy balance, derived from general heat and mass transfer theory (Schymanski et al., 2013; Schymanski and Or, 2016b). For comparison, similar simulations were performed using a simplified model of the leaf energy balance, as described in the appendix of Ball et al. (1988).




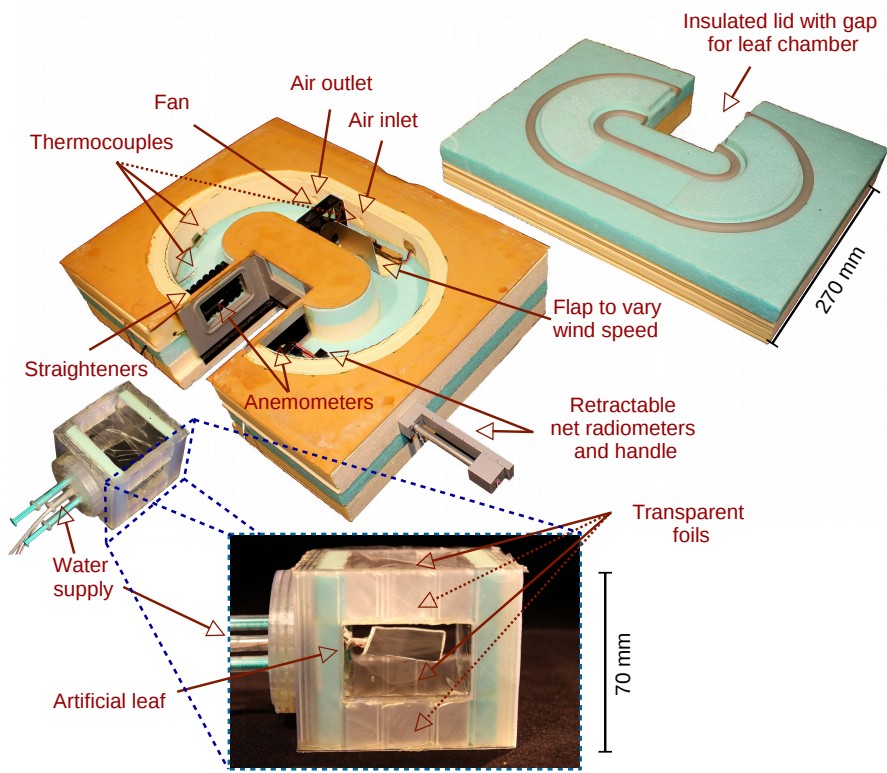

**Figure 2.** Insulated wind tunnel and leaf chamber. The wind tunnel is photographed with its insulated lid removed (top right). The leaf chamber (inset) fits tightly (as air tight as possible) into the empty slot of the wind tunnel on the left. The perspective through the leaf chamber is along the wind flow path (in the upwind direction), illustrating the smooth flow path of 5 by 3 cm cross section. For a detailed dimensions, see Fig. A4. Dashed arrows point to locations of features that cannot be seen in the pictures.

The models mentioned above assume strong thermal coupling between the surface temperatures on both sides of the leaf and therefore equal leaf temperatures on both sides at steady state, resulting in a single energy balance equation:

$$R_s = R_{ll} + H_l + E_l, \tag{1}$$

where $R_s$ is absorbed short-wave radiation, $R_{ll}$ is the net emitted long-wave radiation, i.e. the emitted minus the absorbed, $H_l$ is the sensible heat flux away from the leaf and $E_l$ is the latent heat flux away from the leaf, all in units of W m$^{-2}$.

The net longwave emission is represented by the difference between blackbody radiation at leaf temperature ($T_l$, K) and that at the temperature of the surrounding objects ($T_w$, in our experiments equal to air temperature, $T_a$, K) (Monteith and Unsworth, 2007):

$$R_{ll} = 2\epsilon_l \sigma (T_l^4 - T_w^4), \tag{2}$$




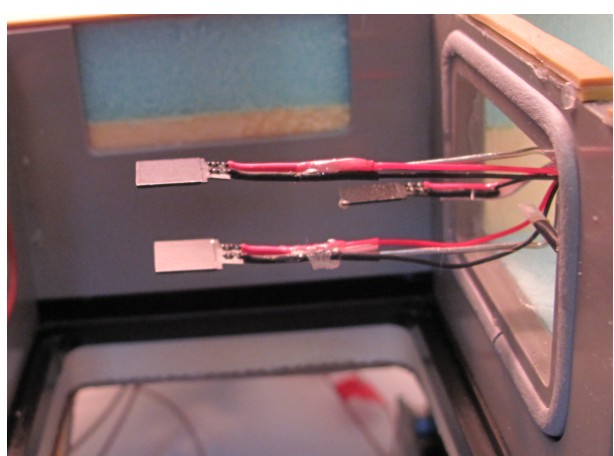

**Figure 3.** Arrangement of net radiometers in measuring position when the leaf chamber is removed. Two sensors are placed 2 cm apart vertically (1 cm above the leaf and 1 cm below the leaf), and one sensor at the level of the artificial leaf, slightly downwind from the leaf.

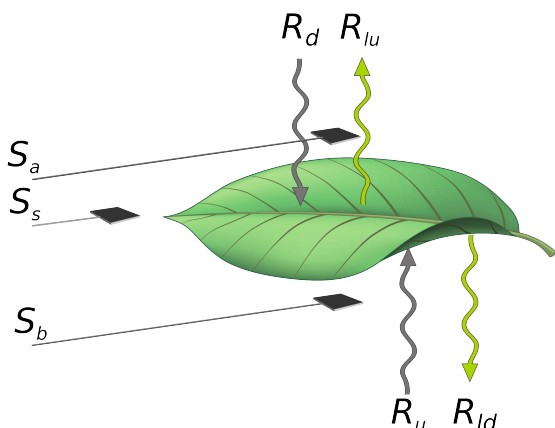

**Figure 4.** Leaf radiative energy exchange components and net radiation sensors in the radiative field. $R_d$: downwelling radiation, $R_{lu}$: longwave radiation emitted upwards by leaf surface, $R_u$: upwelling radiation, $R_{ld}$: longwave radiation emitted downwards by leaf, $S_a$: sensor above leaf, $S_s$: sensor beside leaf, $S_b$: sensor below leaf. See Fig. 3 for a picture of the sensors.





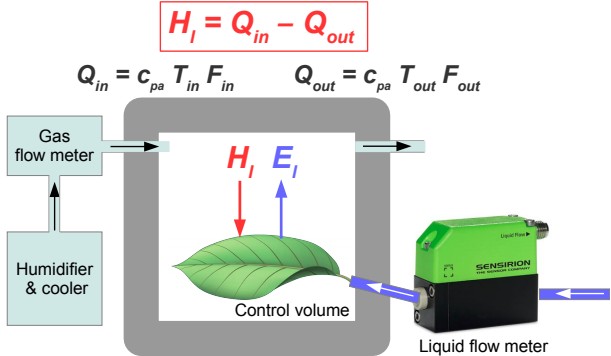

**Figure 5.** Simplified energy balance of insulated wind tunnel. Latent heat flux ($E_l$) is calculated from liquid flow rate into leaf, sensible heat flux ($H_l$) is calculated from difference in heat content of incoming and outoing air ($c_{pa}$: heat capacity of air; $T_{in}, T_{out}$ air temperatures of incoming and outgoing air; $F_{in}, F_{out}$: incoming and outgoing air flow rates).

where $\epsilon_l$ is the leaf's longwave emmissivity ($\approx 1$) and $\sigma$ ($5.67 \times 10^{-8}$ W K$^{-4}$ m$^{-2}$) is the Stefan-Boltzmann constant. Sensible heat flux ($H_l$) is represented as:

$$H_l = 2h_c(T_l - T_a), \tag{3}$$

where $h_c$ (W K$^{-1}$ m$^{-2}$) is the average one-sided convective heat transfer coefficient, determined primarily by leaf size and wind speed.

The latent heat flux is a function of the transpiration rate ($E_{l,mol}$, mol m$^{-2}$ s$^{-1}$):

$$E_l = E_{l,mol} M_w \lambda_E, \tag{4}$$

where $M_w$ (kg mol$^{-1}$ is the molar mass of water and $\lambda_E$ ($2.45 \times 10^{-6}$ J kg$^{-1}$) the latent heat of vaporisation. In the model used here (Schymanski and Or, 2016a), $E_{l,mol}$ is computed as a function of the concentration of water vapour within the leaf ($C_{wl}$, mol m$^{-3}$) and in the free air ($C_{wa}$, mol m$^{-3}$) (Incropera et al., 2006, Eq. 6.8):

$$E_{l,mol} = g_{tw}(C_{wl} - C_{wa}), \tag{5}$$

where $g_{tw}$ (m s$^{-1}$) is the total leaf conductance for water vapour, dependent on diffusive (stomatal) ($g_{sw}$) and aerodynamic (leaf boundary layer) conductance ($g_{bw}$), expressed as follows:

$$g_{tw} = \frac{1}{\frac{1}{g_{sw}} + \frac{1}{g_{bw}}} \tag{6}$$

Note that $g_{sw}$ depends on sizes, shapes and densities of stomata, whereas $g_{bw}$ is a function of leaf size and wind speed.

As an alternative to Eq. 5, $E_{l,mol}$ is commonly expressed as a function of the vapour pressure difference between the free air ($P_{wa}$, Pa) and the leaf ($P_{wl}$, Pa), in which total conductance ($g_{tw,mol}$) is expressed in molar units (mol m$^{-2}$ s$^{-1}$):

$$E_{l,mol} = g_{tw,mol} \frac{P_{wl} - P_{wa}}{P_a} \tag{7}$$





Partitioning of $g_{tw,mol}$ into $g_{sw,mol}$ and $g_{bw,mol}$ is done similarly to Eq. 6. Under a few simplifying assumptions, conductance values can be converted between molar units and [m s$^{-2}$] in the following way (Schymanski and Or, 2016a):

$$g_{sw} = g_{sw,mol} R_{mol} \frac{T_a}{P_a} \tag{8}$$

$R_{ll}$, $H_l$ and $E_l$ (through temperature dependence of $C_{wl}$) depend on leaf temperature ($T_l$) in such a way that for any environmental forcing (irradiance, humidity, temperature and wind speed) and leaf properties (characteristic length scale, leaf diffusive conductance), a steady-state $T_l$ can be found that satisfies the above energy balance equation. For our artificial leaves, the characteristic length scale is 0.03 m, and the leaf diffusive conductance is deduced from foil thickness, as well as sizes and spacings of pores in the laser perforated foils, as described in Appendix A.

The model is explained in detail in Schymanski and Or (2016a), whereas all data, equations and model code necessary to reproduce the results presented here can be accessed online at: https://github.com/schymans/Schymanski_experimental_2016. git

### 2.5.1 Different temperatures on both sides of the leaf

Many leaf gas and energy exchange models assume a single leaf temperature ($T_l$) applicable for both sides of a leaf. This
assumption is justified for very thin leaves or very high leaf thermal conductivities, whereas our infrared images pointed to significant temperature gradients between the wet and the dry side of an artificial leaf (see Section 3.1.2). Hays (1975) measured leaf thermal conductivities ($k_l$) over a range of leaves and found values in the range between 0.27 and 0.57 W m$^{-1}$ K$^{-1}$, compared to the thermal conductivity of air at 0.026 and liquid water at 0.59 W m$^{-1}$ K$^{-1}$. To estimate the potential error introduced by the assumption that both sides of the leaf have the same leaf temperature, we first expressed a conductive heat
flux from upper to lower side of leaf ($Q_l$) as:

$$Q_l = k_l \frac{(T_{l_u} - T_{l_l})}{z_l} \tag{9}$$

where $k_l$ (W m$^{-1}$ K$^{-1}$ is the leaf thermal conductivity, $T_{l_u}$ and $T_{l_l}$ are the upper and lower side of the leaf respectively, while $z_l$ is the thickness of the leaf. We then formulated the energy balance equation for the upper and the lower leaf sides separately, as:

$R_s = E_{l_u} + H_{l_u} + R_{ll_u} + Q_l$ $\hspace{5cm}$ (10)

and

$Q_l = E_{l_l} + H_{l_l} + R_{ll_l},$ $\hspace{5cm}$ (11)

where we assumed that only the upper side of the leaf absorbs shortwave radiation ($R_s$). Eqs. 10 and 11 are equivalent to Eq. 1, with the addition of $Q_l$ and explicit distinction between the two sides, denoted by the subscript $u$ for upper and $l$ for lower
side. Since our model is formulated for forced convection, we assume that the heat transfer coefficient ($h_c$) has the same value on both sides, hence differences in the sensible heat flux are attributed to different leaf surface temperatures only. However $E_{l_u}$





and $E_{l_l}$ can differ due to both different surface temperatures and different stomatal conductances. In the extreme case, e.g. for a hypostomatous leaf, $E_{lu} = 0$, while $E_{ll}$ is calculated similarly to one-sided $E_l$ with $T_l$ replaced by $T_{ll}$. Instead of one equation with one unknown (Eq. 1 with unknown $T_l$), we now obtain two equations with two unknowns (Eqs. 10 and 11 with $T_{l_u}$ and $T_{l_l}$), as all variables in these two equations are functions of only measured quantities as well as $T_{l_l}$ and/or $T_{l_u}$. The equations were solved numerically using the open source software SageMath (Stein and et al., 2016), and the code is available online at https://github.com/schymans/Schymanski_experimental_2016.git. In the results section, we compare measured fluxes and leaf temperatures with simulated values using both the uniform temperature model ("bulk") and the model based on different surface temperatures on both leaf sides ("2s").

## 3   Results

### 3.1   Characterisation of artificial leaves

#### 3.1.1   Pore properties and stomatal resistances

The laser-perforated aluminium foils have a shiny and a matte side, and confocal laser scanning microscope (CLSM) images of the perforated foils (Fig. 6) were taken on the matte side, prior to construction of the artificial leaves. The matte side was facing outwards after construction of the artificial leaves. More detailed analysis of pore geometries was performed on duplicate foils and suggests that the laser perforations were done from the shiny side, resulting in irregular surfaces around the pores and slightly conical pore geometries with smaller diameters on the matte compared to the shiny side. Therefore, images taken on the matte side may result in under-estimation of the effective pore sizes. When we compared estimations of pore sizes and conductances based on images taken on either side of the aluminium foil, we found higher conductance values by up to 50% if images were taken on the shiny side compared to the matte side (Fig. A1, Table A1). Detailed analysis of individual pore geometries also revealed that the average cross-sectional pore area over the typical 25 $\mu$m pore length could be up to 50% larger than the areas measured 10 $\mu$m below the foil surface (Appendix A). To account for all these uncertainties and potential biases, Table 1 provides ranges of values deduced from at least three images on each side of a foil (columns 1–4) and a column with stomatal conductance ($g_{sw}$) values resulting from the assumption that the average cross-sectional pore areas were 50% larger than deduced from the images (Column 5). The last column in Tab. 1 represents $g_{sw}$ values deduced from wind tunnel experiments described in Section 3.2, which were remarkably consistent with the theoretical values in Column 5.

The perforated foils were glued to the artifical leaf along the edges (Fig. 1, while they only adhered to the wet filter paper by capillary forces if a water film was present between the filter paper and the perforated foil. When carefully saturating an artificial leaf, we found that water could be held within the pores (Fig. A3, which would result in a dramatic shortening of the diffusive path length across the pores, from 25 $\mu$m (foil thickness) to less then 1 $\mu$m if there were no considerable head loss along the flow path, as the water reservoir was kept only a few centimetres below the leaf to reduce the risk of embolism. To get an appreciation for the maximum effect of capillary rise within the pores on stomatal resistance, we derived stomatal conductance values for the different perforated foils based on both 25 and 0 $\mu$m pore length, and found that a reduction of





**Table 1.** Perforation characteristics and resulting diffusive conductances ($g_{sw}$, from Eq. A3), either taken the original pore area deduced from CLSM images or 50% increased pore area, taking into account conical shapes of pores. In the last column, we provide effective stomatal conductances deduced from wind tunnel experiments.

| Pore density mm$^{-2}$ | Pore area $\mu$m$^{-2}$ | Pore radius $\mu$m | $g_{sw}$ m s$^{-1}$ | $g_{sw}$ (1.5×area) m s$^{-1}$ | $g_{sw}$ (wind tunnel) m s$^{-1}$ |
|---|---|---|---|---|---|
| 52–68.8 | 859–1240 | 16–20 | 0.032–0.052 | 0.046–0.076 | 0.05 |
| 27.3–38.2 | 710–1572 | 15–22 | 0.015–0.032 | 0.022–0.046 | 0.035–0.042 |
| 7.1–7.8 | 890–1886 | 16–24 | 0.004–0.009 | 0.006–0.012 | 0.007–0.009 |

$g_{sw}$: diffusive (stomatal) conductance for water vapour; $d_p$: pore depth.

the pore length from 25 to 0 $\mu$m could result in a three-fold increase in estimated stomatal conductance (data not shown). However, as presented in Table 1, the stomatal conductance values deduced from wind tunnel experiments are more consistent with values determined under the assumption that no water was held in the pores.

### 3.1.2 Leaf thermal mapping

We placed a thick artificial leaf (0.2 mm thick filter paper with 0.2 mm thick thermocouple fed in from the side) and a thin artificial leaf (0.1 mm thick membrane filter and 0.13 mm thick thermocouple) under the thermal IR camera and took images of their surface temperatures. Temperatures increased from the leading edge downwind by no more than 1.1 K (Fig. 7). Interestingly, surface temperatures of the wet surfaces were lower wherever it was detached from the underlying aluminium tape, e.g. along the thermocouple wires and air pockets (Figs. 7 and 8). We also found that the surface temperature of the dry side of the leaf was warmer by up to 1.4 K compared to the wet side (Fig. 8). Please refer to the discussion section (4.3) for the relevance of these findings.

### 3.2 Leaf wind tunnel experiments

Experiments were performed using artificial leaves with different perforation densities under varying air humidity or varying wind speed, with and without shortwave radiation. In addition to the artificial leaves with pore densities given in Table 1, we also used artificial leaves without a perforated foil, i.e. with a wet surface on the lower side, producing non-restricted one-sided leaf boundary layer transfer of water vapour. For simulated energy balance components and leaf temperatures, we chose diffusive conductance values ($g_{sw}$) that best matched the observed transpiration rates and then compared these with conductance values computed from laser perforation analysis (Table 1). Simulations were performed using the original model assuming equal leaf temperatures on both sides of the leaf (bulk) and the two-sided leaf temperature model (2s). We also adjusted leaf thermal conductivities ($k_l$) within the range between air and water, to best reproduce measured leaf temperatures in the 2s model. Figs. 9 and A6–A7 represent experiments in the absence of shortwave irradiance, and their 5 panels include (from top to bottom): latent and sensible heat flux, sums of latent and sensible heat flux along with net absorbed radiation, leaf-air temperature





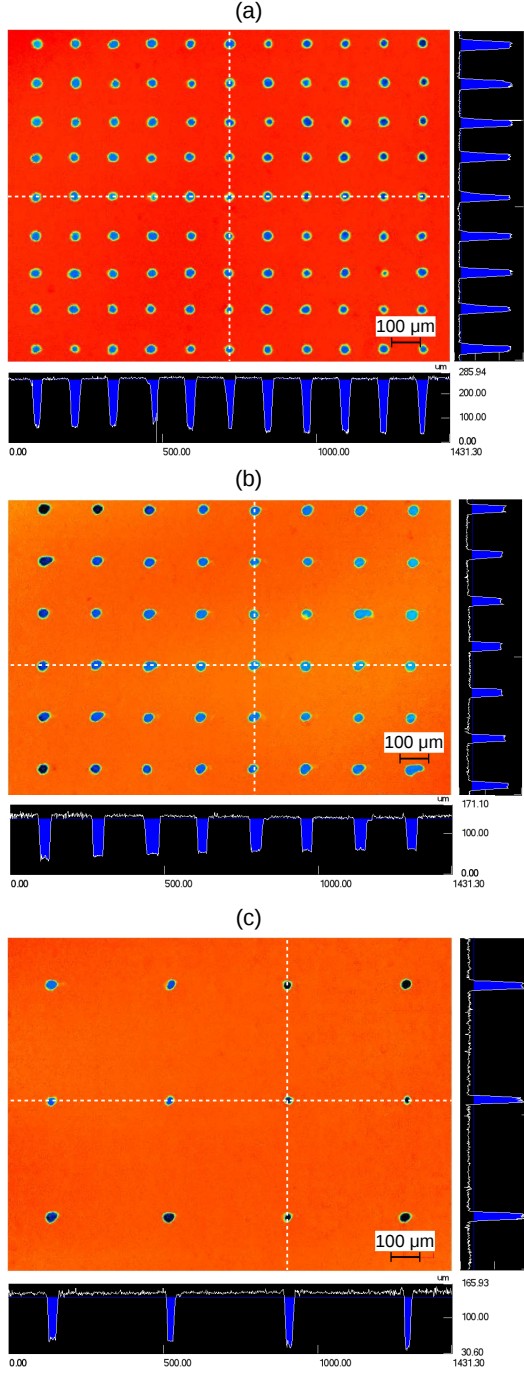

**Figure 6.** Example confocal laser scanning microscope (CLSM) images of perforated foils summarised in Tab. 1. (a) 64 perforations per $mm^2$, (b) 35 perforations per $mm^2$, (c) 7.8 perforations per $mm^2$. Black bars at the bottom and on the right of each picture show topographic profiles of transects crossing perforations (white dashed lines in main images), with the detection thresholds marked as blue filled areas.





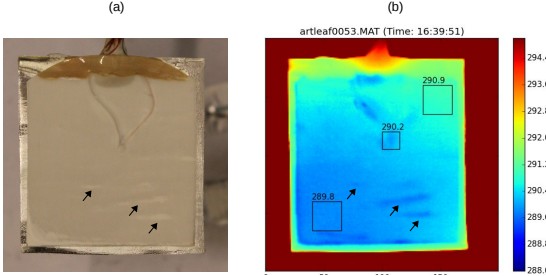

**Figure 7.** Wet side of thin artificial leaf at 3 m s$^{-1}$ wind speed. (a) Photographic image, (b) infrared temperature map, with average temperatures in different sub-areas. Arrows indicate air pockets between wet filter paper and underlying aluminium tape. Wind direction is from bottom to top of the images. Colour bar indicates temperatures in K.

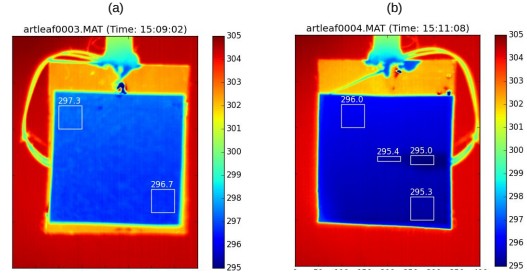

**Figure 8.** Infrared temperature maps of thick artificial leaf in the absence of wind. (a) dry side, (b) wet side. Colour bar indicates temperatures in K, while white labels above white rectancles indicate average temperature (in K) over the respective rectangles. Note that the wet side is significantly cooler than the dry side of the leaf.

difference, leaf conductance to water vapour and convective heat transfer coefficient. For the latter two, observed values were deduced from observed fluxes and leaf-air temperature and vapour concentration gradients. Fig. 10 represents experiments under irradiance. Since sensible heat flux could not be measured accurately under irradiance (see discussion, Section 4.1), we left out the bottom panel. As seen in Fig. 10a, the over-estimation of observed sensible heat flux ($H_l$, empty circles), was likely

in the order of 500 W m$^{-2}$, which also led to a mismatch in the energy balance by a similar amount (second panel from top).

In the absence of shortwave radiation, both sensible and latent heat fluxes were very consistent between observations and model simulations (top panels in plots), no matter whether vapour pressure or wind speed was varied. The sums of observed latent and sensible heat flux varied between 20 and 120 W m$^{-2}$ and were largely consistent with simulated radiative exchange of the leaf in half of the cases, while exceeding the simulated exchange of radiative energy in the other half of the experiments. Our

net radiation sensors were not able to confirm such high radiative energy exchange rates, and generally under-estimated the net absorbed longwave radiation by more than half, compared to simulations (red dots in Figs. 9 (b), 10 and A6 (b). The observed leaf temperatures were also generally under-estimated by the bulk model (absolute leaf-air temperature difference was over-



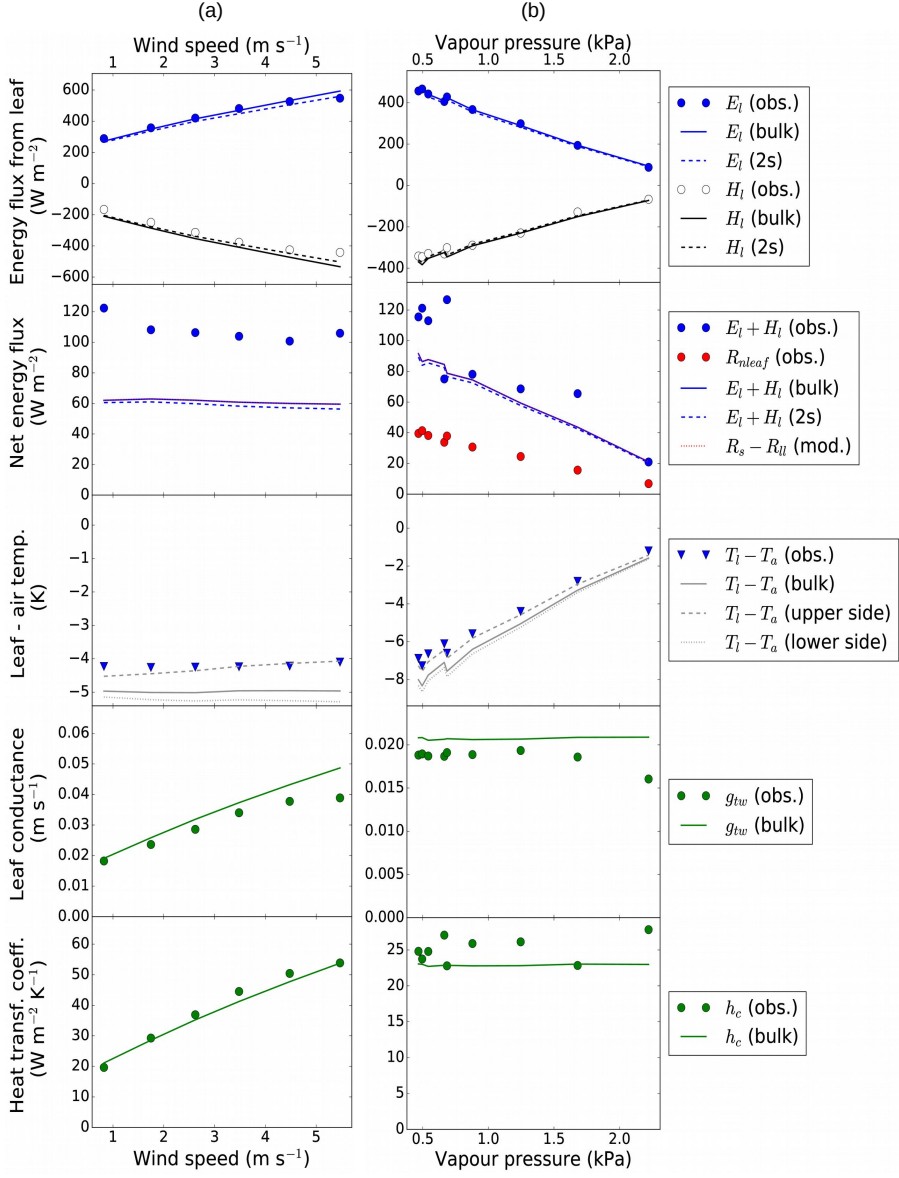

**Figure 9.** Artificial leaf with wet surface on the lower side (no stomatal resistance), under (a) varying wind speed and (b) varying vapour pressure. Numerical model results (lines) based on the same boundary conditions as observations (symbols). $E_l$: latent heat flux; $H_l$: sensible heat flux; $R_s - R_{ll}$: absorbed net radiation; $T_l - T_a$: leaf-air temperature difference; $g_{tw}$: total leaf conductance to water vapour; $h_c$: convective heat transfer coefficient; "bulk": bulk leaf temperature model; "2s": model based on different leaf temperatures on both leaf sides. Boundary conditions: $g_{sw} = 999$ m s$^{-1}$; $R_s = 0$; $T_a = 295.4 - 295.6$ K (a) and $295.4 - 296.6$ (b); $P_{wa} = 1200 - 1342$ Pa (a); $v_w = 1.0$ m s$^{-1}$ (b); $k_l = 0.1$ W K$^{-1}$ m$^{-1}$; $z_l = 0.5$ mm.





estimated by 0.5–1 K). However, when solving the leaf energy balance for each leaf side separately (considering conductive heat transport across the leaf towards the transpiring side, Section 2.5.1), the observed leaf temperatures were consistent with

the simulated leaf temperatures of the non-transpiring side of the leaf (Figs. 9, 10 (a), A6 and A7). Interestingly, solving for the temperature gradient between the two sides of the leaf did not have much effect on the simulated heat fluxes (top panels in the plots). Note that the values of $k_l$, chosen to reproduce observed leaf temperatures, varied between experiments, between 0.03 and 0.3 W K$^{-1}$ m$^{-1}$, compared to values of 0.026 for air and 0.59 for water.

Irradiation of the leaf with 370–550 W m$^{-2}$ shortwave radiation resulted in large over-estimation of sensible heat flux in

the observations and hence unrealistically high sums of latent and sensible heat fluxes, while simulated and latent heat fluxes reproduced the observed very accurately (top panels in Fig. 10). Observed leaf temperatures were still higher than the simulated, while the observed radiative exchange was relatively close to simulated $R_s - R_{ll}$ (second panels in Fig. 10 (a) and (b)). Note that the radiation sensor placed downwind of the leaf (Fig. 3) did not produce reliable radiation values, as the readings were affected by leaf temperature (data not shown). For computing $R_{nleaf} = R_s - R_{ll}$, we hence subtracted the reading of the sensor

placed below the leaf from the reading of the sensor placed above the leaf.

## 4  Discussion

### 4.1  Measurement of leaf energy balance components

Our experimental setup enables independent measurement of leaf-scale exchange of latent and sensible heat in the absence of shortwave irradiance. This was confirmed on a variety of artificial leaves with different diffusive conductances, under varying

vapour pressure and wind speed. Energy balance closure in the absence of light was generally within 60 W m$^{-2}$ s$^{-1}$, as illustrated in the net energy exchange panels in Figs. 9, A6, and A7.

Under shortwave irradiance, however, sensible heat flux deduced from measurements was largely over-estimated, probably due to absorption of radiation by surfaces within the wind tunnel, despite coating with reflective tape and a second transparent window below the leaf. It is also important to note that, despite construction of the wind tunnel using thermally insulating

materials, the internal air temperature had to be kept close to lab air temperature, in order to prevent conductive heat exchange across the chamber walls. We found that a temperature difference of only 2 K between the air within the wind tunnel and outside could result in a bias in estimated sensible heat flux by 300 W m$^{-2}$ in our experimental setup (data not shown).

The exchange of longwave radiation between the leaf and the surroundings was not captured in a consistent way by our experimental setup, suggesting that the measurements systematically under-estimated longwave radiation away from the leaf

by more than 50%. Consideration of the viewing angle of the net radiation sensor would only correct the estimates by 20% (Section B3). The reason for the bias is most likely that the miniature radiation sensors were calibrated against an industrial net radiometer using shortwave radiation as main energy source, while their capability to absorb in the longwave range was not tested. In the presence of shortwave radiation, the sensors were adequate to characterise the radiative load on the leaf, as illustrated by the correct simulation of latent heat flux in Fig. 10. Note, however, that the sensor placed at leaf level, but slightly

downwind from the leaf, did not return reliable values, as it was affected by leaf temperature (data not shown). This is likely





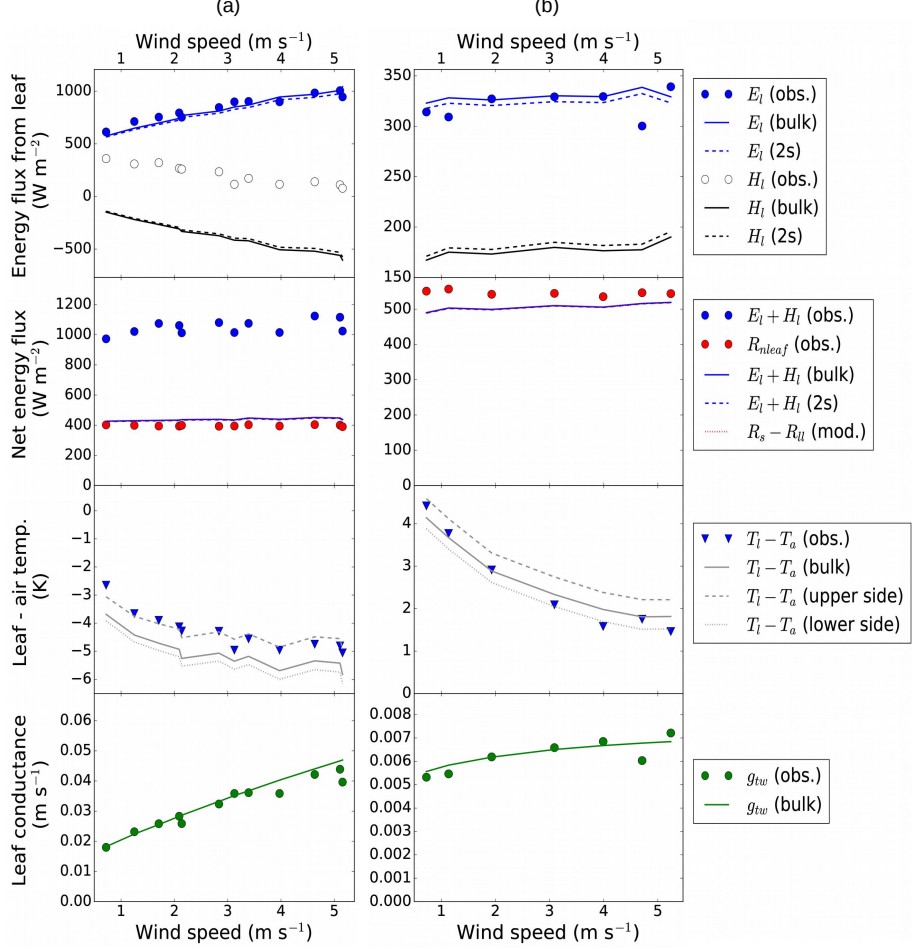

**Figure 10.** Artificial leaves with (a) wet surface on the lower side and (b) 7 perforations mm$^{-2}$. Numerical model results (lines) based on the same boundary conditions as observations (symbols). $E_l$: latent heat flux; $H_l$: sensible heat flux; $R_s - R_{ll}$: absorbed net radiation; $T_l - T_a$: leaf-air temperature difference; $g_{tw}$: total leaf conductance to water vapour; $h_c$: convective heat transfer coefficient; "bulk": bulk leaf temperature model; "2s": model based on different leaf temperatures on both leaf sides. Boundary conditions: $g_{sw} = 999$ (a) and 0.008 (b) m s$^{-1}$; $R_s = 370-380$ W m$^{-2}$ (a) and 530–550 W m$^{-2}$ (b); $T_a = 295.8 - 296.6$ K (a) and $295.4 - 296.6$ K (b); $P_{wa} = 521-802$ Pa (a) and 341–356 Pa (b); $k_l = 0.27$ W K$^{-1}$ m$^{-1}$ (a) and 0.3 (b) W K$^{-1}$ m$^{-1}$; $z_l = 0.5$ mm.

due to vertical temperature gradients in the aerodynamic wake of the leaf, which could result in compensatory heat flux through the sensor in addition to that caused by absorbed radiation. The sensors 1 cm above and below the leaf surface were unaffected by the leaf boundary layer and produced a signal that was weakly affected by the leaf temperature, consistent with the leaf temperature effect on the net emission of longwave radiation.





### 4.2 Utility of artificial leaves

Despite many inherent limitations in mimicking real leaves, the artificial leaves proved very useful for analysis of steady state leaf energy balance components under constant leaf properties, in particular leaf diffusive conductance. This is supported by the reproduction of experimental results using a model with constant stomatal conductance ($g_{sw}$), both for varying vapour pressure and wind speed. Water supply to the evaporating sites via a flow-monitored tube and porous filter paper resulted in relatively homogeneous conditions over the leaf surfaces, as evidenced by our infrared images (Figs. 7 and 8). However, in some cases, water transport to the edges of the artificial leaves ceased over time, which could be seen at the end of the experiment as dry patches on the filter paper. The bottom left corner of the leaf in Fig. 7 shows the onset of such an effect. To detect this effect in artificial leaves where the filter paper was covered by a laser-perforated foil, experiments were run in two directions, first increasing wind or vapour pressure and then decreasing it again. Whenever we found a clear reduction in transpiration at the end of an experiment compared to the start, we discarded the whole data set. We kept the free water surface of the water supply tank only a few cm below the position of the leaf in order to facilitate water transport all the way to the edges of the filter paper and avoid hydraulic failure.

One of our aims was to produce evaporating surfaces with a diffusive conductance ($g_{sw}$) that is known *a priori*. However, the uncertainty in the computation of $g_{sw}$ based on CLSM topographical images of our laser-perforated foils was substantial (Tables 1 and A1). Roughly 25% uncertainty was introduced by irregularities in pore sizes, resulting in different average pore areas in different images of the same foil (see ranges in the fourth column of Tab. 1). An additional potential bias by 30% was caused by the conical shape of the pores. As shown in Fig. A2, the mean diameter of a pore across the whole pore length was likely 25% larger than the diameter measured at 10 $\mu$m distance from the surface. This would result to 50% larger mean cross-sectional areas in the pores and 30–50% larger values for $g_{sw}$, as shown in the fifth column of Tab. 1.

We also found that a substantial uncertainty could arise from lack of knowledge about the position of the evaporating sites if they were within the perforations. As illustrated in Fig. A3, water could have entered the pores and been drawn up to the surface, reducing the diffusion distance through the pores. In the extreme case, this could have led to a three-fold increase in $g_{sw}$ compared to assuming that the evaporating sites are below the pores. However, although this was observed after careful saturation of an artificial leaf under the CLSM, the $g_{sw}$ values deduced from wind tunnel measurements (last column in Table 1) were consistent with those calculated under the assumption that the evaporating sites were below the pores. Since emptying of previously water-filled pores is irreversible as long as the hydraulic head in the artificial leaves is negative, we expect most pores to be air-filled in our experiments.

### 4.3 Leaf temperatures

Despite the thin leaf design, our data suggest that significant temperature gradients can occur between the dry and the evaporating side of the leaf. This was confirmed directly by the infrared images of the wet and dry leaf surfaces (Fig. 8), and by the consistent bias in leaf temperature simulated by the bulk model compared to observed leaf temperatures. Note that the dry leaf surface was black painted aluminium tape, whereas the wet side was paper, which could result in emissivity differences and





hence bias in the surface temperature differences deduced from infrared imaging. However, given that the infrared emissivity of a wet surface is close to 1, while that of untreated aluminium is very low, we would expect any bias caused by emissivity

differences to result in under-estimation of the dry surface temperature, meaning that the temperature difference could even be higher than 1.4 K we found.

The level of de-coupling between wet and dry side of the leaf depends on the leaf thermal conductivity, as illustrated by the cooler surface temperatures wherever little air intrusions between the wet filter paper and dry aluminium tape occurred (Fig. 7). Since the thermocouples within the leaf were in contact with the upper aluminium tape of the leaf, they most likely

are strongly coupled to the surface temperature of the dry side of the leaf. This may explain the persistent positive bias of the thermocouple reading compared to model simulations, even if the model simulations reproduced the leaf energy balance components very well. However, the magnitude of the bias is not always explained by decoupling between the two leaf sides, as the corresponding leaf thermal conductance would have to be near that of air in some simulations. More targeted experiments are needed to rule out a systematic error in the representation of the sensitivity of transpiration to leaf temperature.

It is remarkable that consideration of conductive heat flux through the leaf in the 2s-model has a strong effect on simulated leaf temperatures, but not so much on the simulated latent and sensible heat fluxes, compared to the bulk leaf temperature model. This is likely because the simulated bulk leaf temperature is between the temperatures of the transpiring and the dry side in our hypostomatous leaf replica. Under-estimation of sensible heat flux on one side is hence partly compensated by over-estimation of sensible heat flux on the other side. Consideration of conductive heat flux through the leaf increases leaf

temperature on the dry side because of decoupling from evaporative cooling on the wet side. On the wet side, the decoupling reduces heat input from the dry side, but if the leaf is colder than the air, this is partly compensated for by increased uptake of sensible heat as the wet side of the leaf cools. This has the result that the leaf temperature on the wet side is closer to the simulated bulk leaf temperature than the leaf temperature on the dry side, resulting in little difference in latent heat flux between the bulk and the 2s-model.

Given that significant temperature differences between the dry and evaporating sides of the leaf were simulated even for leaf thermal conductance values similar to natural leaves (up to 1 K in Fig. 10), we conclude that care must be taken when inferring leaf-internal vapour pressure from leaf temperature measurements. However, for the simulation of latent and sensible heat fluxes, a bulk formulation seems adequate due to the compensating effects of under-estimating leaf temperature on one side and over-estimating on the other.

## 4.4   Leaf conductances deduced from experiments

By inverting Equations 3 and 5, we computed the effective one-sided convective heat transfer coefficients ($h_c$) and total leaf conductances to water vapour ($g_{tw,mol}$) based on observed latent and sensible heat fluxes, air and leaf temperatures and air vapour pressure. These are reproduced in Panels (d) and (e) of the results plots, and compared with theoretical values based on wind speed and leaf properties. Both theoretical and inferred values were reasonably close to each other and responded

to wind speed in a consistent way. However, the theoretical values for $g_{tw}$ were consistently higher than the deduced values,





while the theoretical values for $h_c$ were consistently lower than the deduced. This can be explained in the context of biased leaf temperature measurements:

Observed and simulated leaf temperatures generally increased with increasing wind speed in the absence of shortwave radiation, except for artificial leaves without perforated foils (i.e. lower wet surface exposed to air), where leaf temperature did not change with wind speed. Consistent with the higher observed leaf temperatures relative to simulations, leaf conductance to water vapour, calculated from observed leaf temperature, vapour pressure and latent heat flux, was generally lower than the simulated conductances. For the same reason, convective heat transfer coefficients computed from observed sensible heat flux and leaf temperature were generally higher than simulated values. This is because higher leaf temperature implies an increased gradient for latent heat flux and reduced gradient for sensible heat flux when leaf temperatures are below ambient. Note that this feedback has recently also been shown to result in reduced transpiration and/or increased leaf water use efficiency with increasing wind speed when irradiated leaves are warmer than ambient air (Schymanski and Or, 2016b).

### 4.5 Potential for new insights and limitations

Independent measurement of sensible and latent heat fluxes from artificial leaves with fixed stomatal conductance offers various opportunities to test our understanding of the physics of leaf gas and energy exchange. Our experiments have already enabled discovery of inconsistencies in the widely used Penman-Monteith equation for transpiration (Schymanski and Or, 2016a) and surprisingly strong temperature gradients between two sides of a hypostomatous leaf (this study). The experimental setup presented here could also be used for detailed studies of the role of stomata sizes, shapes and arrangements for leaf gas and energy exchange as well as isotope partitioning. Due to the explicit control volume approach with thermal insulation, the setup could also be used to study physical components of surface-atmosphere feedbacks at the laboratory scale.

However, the estimation of sensible heat flux from the chamber energy balance requires steady-state conditions, where heat exchange between the chamber air and the chamber walls are negligible. In our experience, it takes tens of minutes to hours before a steady state is achieved after an experimental change in the boundary conditions. This is due to the low heat capacity of the chamber air compared to that of the wind tunnel walls and any equipment placed within the wind tunnel, and makes the setup of limited use for living leaves that vary their stomatal conductance at time scales shorter than the characteristic time scale of the whole chamber. Furthermore, for the estimation of sensible heat flux from an irradiated leaf, it would be necessary to focus a light beam on the leaf surface only and avoid any absorption of stray light by other surfaces inside the wind tunnel. And finally, in order to close the energy balance, it would be desirable to have reliable measurements of the leaf's radiative exchange, including longwave radiation. It may be necessary to develop new sensors that are small enough to fit into the wind tunnel without modifying air flow and the energy balance of the wind tunnel.

## 5 Conclusions

The experimental setup presented here allows for the first time simultaneous and independent measurement of gas and energy exchange by artificial leaves under fully controlled conditions. In the absence of shortwave irradiance, the results from the





experimental setup were remarkably consistent with theoretical predictions for latent and sensible heat fluxes. The experiments presented here also highlight some unexpected difficulties in characterising leaf temperature due to strong temperature gradients

between dry and evaporating leaf sides of planar leaves. More development is needed to achieve reliable measurement of the leaf's radiative energy exchange and sensible heat flux of an irradiated leaf. Preliminary experiments with irradiated leaves suggest that issues related to light absorption by internal windtunnel surfaces need to be resolved. Additionally, we plan experiments using live leaves to test the energy balance under a range of boundary conditions and effects of leaf shape and surface properties on the partitioning between sensible and latent heat flux. This is not possible using common state-of-the-art

leaf gas exchange systems, as they do not permit characterisation of the radiative energy load (net radiation) and exchange of sensible heat by the leaf.

## 6   Code and data availability

All code and data used to generate the results presented in this paper is available online at https://github.com/schymans/ Schymanski_experimental_2016.git.





**Appendix A: Calculation of diffusive leaf conductance from pore dimensions**

Diffusive conductance ($g_{sw}$) for the perforated foils was computed based on pore sizes and densities, following the derivations by Bange (1953), as summarised by Lehmann and Or (2015). We assumed that the stomatal conductance results from two resistances in series, the throat resistance ($r_{sp}$), dependent on the areas of the pores and the thickness of the perforated foil ($d_p$), and the vapour shell resistance ($r_{vs}$), dependent on the size and spacing of the stomata, which can be understood as the

resistance related to distribution of the point source water vapour over the entire one-sided leaf boundary layer. We hereby neglect any internal resistance (termed "end correction" by Lehmann and Or, 2015), as we assume that the wet filter paper has direct contact with the perforated foil. The throat resistance ($r_{sp}$, m$^2$ s mol$^{-1}$) was computed as (Eq. 1 in Lehmann and Or, 2015):

$$r_{sp} = \frac{d_p}{A_p k_{dv} n_p} \tag{A1}$$

where $k_{dv}$ is the ratio of the vapour diffusion coefficient and the molar volume of air ($D_{va}/V_m$), and $A_p = \pi r_p^2$. For the vapour shell resistance ($r_{vs}$, m$^2$ s mol$^{-1}$), we use the formulation originally proposed by Bange (1953):

$$r_{vs} = \left( \frac{1}{4 r_p} - \frac{1}{\pi s_p} \right) \frac{1}{k_{dv} n_p} \tag{A2}$$

where $s_p$ (m) is the spacing between stomata, inferred from the images as $s_p = 1/\sqrt{n_p}$. Stomatal conductance ($g_{swmol}$, mol m$^{-2}$ s$^{-1}$) was then calculated as:

$$g_{sw,mol} = 1/(r_{sp} + r_{vs}), \tag{A3}$$

and conversion to units of $g_{sw}$ (m s$^{-1}$) was done following Schymanski and Or (2016a), i.e. $g_{sw} = g_{sw,mol} R_{mol} T_a / P_a$.

At least three confocal laser scanning microscopy (CLSM) images of each perforated foil were examined and perforations were identified based on the 3D topography, using an elevation threshold of 10 $\mu$m below the median to define a pore (Figs. 6, A2). Average pore area ($A_p$, m$^2$) and number of pores per surface area ($n_p$, m$^{-2}$) were computed for each image using the

proprietary Keyence VK Analyzer software, ver. 3.3.0.0. Pore radius ($r_p$, m) was deduced from average pore area assuming circular pores, while pore spacing ($s_p$, m) was computed based on the assumption of regular pore spacing as $s_p = 1/\sqrt{n_p}$. The pore depth ($d_p$, m) was assumed to be the same as the foil thickness of 25 $\mu$m. To assess the effect of water films entering the pores (Fig. A3), calculations were made for both $d_p = 25$ and $d_p = 0$ $\mu$m. Note that the foils had a shiny and a matte side and laser cutting of the pores was performed from the shiny side, whereas the foils were mounted on the artificial leaves with the

matte side exposed to the air. Due to the procedure of laser cutting, the surface surrounding the pores was not smooth on the shiny side, but featured ridges and a larger indent around the pores. This could result in significantly greater pore sizes when the same image analysis was done on images taken on the shiny side (Fig. A1 and Table A1). To assess how representative the pore cross-sectional areas 10 $\mu$m away from the surface were of the average pore cross-sectional areas, we chose a single pore in the same foil as in Fig. A1, scanned it from both sides at a higher magnification and aligned the pore profiles in a

way to represent a cross-section along the entire pore length (Fig. A2). We found that pore sizes determined at 10 $\mu$m depth





**Table A1.** Perforation characteristics and resulting stomatal conductances for a foil with 7.8 pores mm$^{-2}$, scanned three times on each side (see example scans in Fig. **??**). For each image, an average pore size and pore density was computed, which was then used to compute stomatal conductance ($g_{sw}$, assuming that pore length ($d_p$) equals foil thickness of 25 $\mu$m. Ranges given in the table represent the ranges obtained from three images on each side.

| | Pore area $\mu\mathrm{m}^{-2}$ | Pore radius $\mu\mathrm{m}$ | $g_{sw}$ ($d_p = 25\mu\mathrm{m}$) m s$^{-1}$ |
|---|---|---|---|
| Matte side | 890–1231 | 17–20 | 0.006–0.008 |
| Shiny side | 1376–1886 | 21–24 | 0.009–0.011 |

$g_{sw}$: diffusive conductance from Eq. A3; $d_p$: pore depth.

represented the sizes of the throats of the pores and might under-estimate the mean cross-sectional pore areas when averaged along the pore axis by up to 50%. Therefore, we computed values of $g_{sw}$ based on the original pore areas determined using the procedure described above (Columns 1–4 in Table 1) and additionally did the same calculations assuming 50% larger pores (Column 5 in Table 1). The latter were most consistent with conductance values deduced from leaf wind tunnel experiments 460 (last column in Table 1).




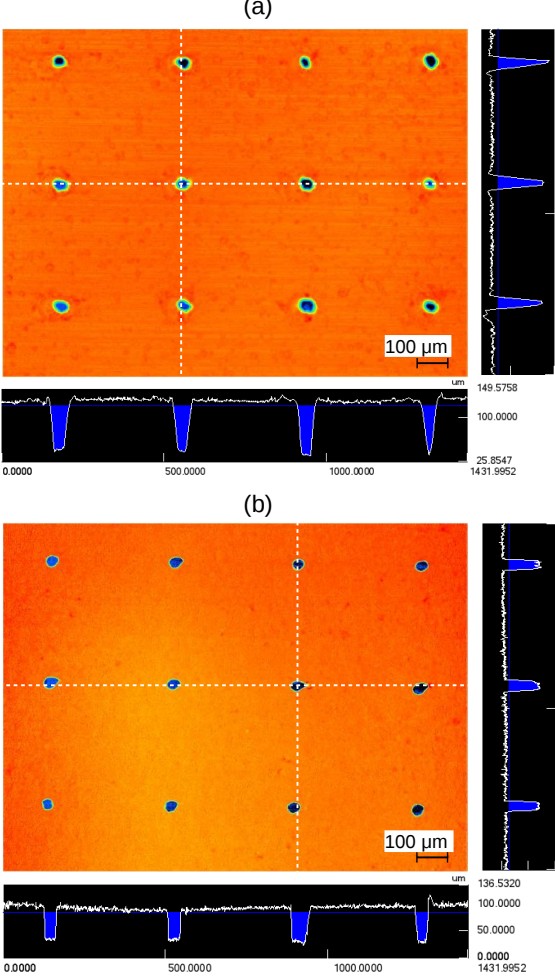

**Figure A1.** Perforations of the same foil (7 pores mm$^2$), scanned on the shiny side (a) and on the matte side (b). Colours represent surface elevation as shown in the profile below and to the right of each panel. Ranges of pore sizes and densities are given in Tab. A1.





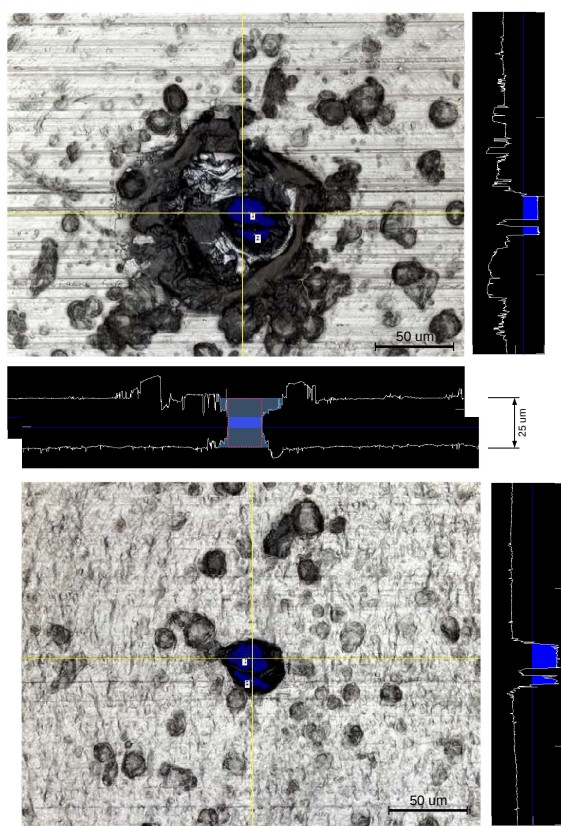

**Figure A2.** Single pore measured from both directions (top: shiny side; bottom: matte side) for the same foil as in Fig. A1. Black bars illustrate measured height profiles across the images, corresponding to the transects indicated by yellow lines. The horizontal profiles are shown in the middle, with the profile belonging to the bottom image mirrored and aligned to produce an estimated cross-section of the foil and the pore, in combination with the height profile of the upper image. The red dashed box represents the hypothetical pore profile deduced from pore diameter 10 $\mu$m below the surface, whereas the light blue shaded area represents the detailed profile of the pore. The shaded area is > 20% larger than the area of the red box, indicating that the average pore diameter is > 20% larger and the average cross-sectional area 40-50% larger than that deduced from the lower image alone.



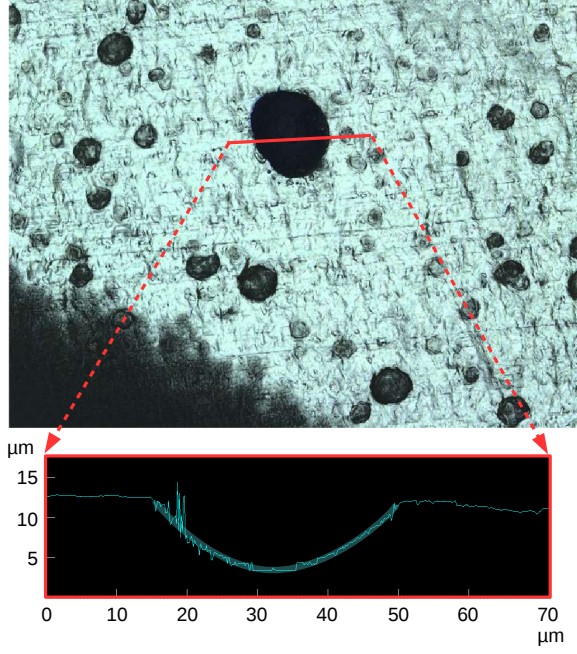

**Figure A3.** Profile of water meniscus in artificial pore at -24.4 cm hydraulic head. A leaf with a pore diameter of 35 $\mu$m and very low pore density (1.8 pores mm$^{-2}$ was used to observe the position of the water meniscus under suction in a single pore as the hydraulic head was varied by progressively lowering the water supply reservoir below the level of the artificial leaf. Top: laser scanning image of the pore with a transect marked across the pore. Note that the smaller black patches are not pores, but surface dents on the surface of the aluminium foil. Bottom: Magnified height profile of the section depicted in the top figure, depicting the surface of the water meniscus inside the pore. For clarity, the supposed water surface is enhanced by a thicker semi-transparent blue line.

## Appendix B: Details on leaf wind tunnel and computations

The leaf wind tunnel is described in the main text (for detailed dimensions, see Fig. A4), while here we describe the calculations to deduce the quantities of interest from measured quantities.

### B1   Inference of sensible heat flux from wind tunnel measurements

Sensible heat exchange between the air and the leaf was inferred from steady state chamber heat balance, based on the following assumptions:

1. Heat conduction through the wind tunnel walls is negligible.

2. Heat input by fan is equal to its electric power consumption.

3. The molar outflow of dry air equals the molar inflow of dry air, while the molar outflow of water vapour equals the molar
inflow plus the evaporation rate.




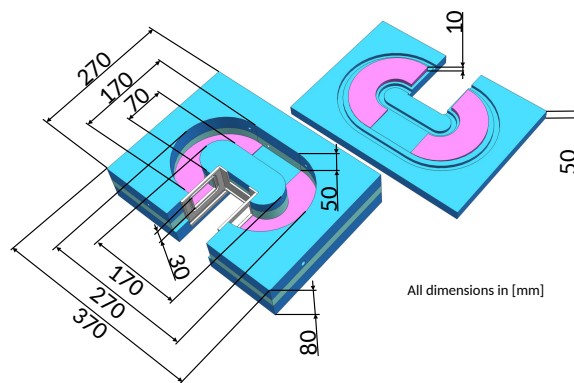

All dimensions in [mm]

**Figure A4.** Wind tunnel dimensions. At the back end, the cross section is 5 cm high, to accommodate the fan. The cross-section is gradually reduced to 3 cm at the transition to the leaf chamber. See also Fig. 2. The ground area of the circular tunnel is 288 cm$^2$, which, assuming an average height of 4 cm, results in 1.152 l air volume in the chamber. The circumference of the wind tunnel is 73.4 cm, resulting in a conductive air-wall interfacial area (neglecting the inner wall) of $2 \times 288\text{cm}^2 + 73\text{cm} \times 4\text{cm} = 868\text{cm}^2$.

4. The heat content of the incoming air can be calculated from its flow rate, humidity and temperature, measured inside the duct through the wind tunnel wall.

5. The heat content of the outgoing air can be calculated from its flow rate, average humidity and temperature.

Based on these assumptions, the energy balance of the wind tunnel is written as:

$$0 = L_A H_l + Q_{in}$$
$$+ (F_{in,mol,a} M_{air} c_{pa} + F_{in,mol,w} M_w c_{pv}) T_{in}$$
$$- (F_{out,mol,a} M_{air} c_{pa} + F_{out,mol,w} M_w c_{pv}) T_{out} \tag{B1}$$

where $L_A$ is the leaf area (m$^2$), $Q_{in}$ is the heat input by the fan (W), $F_{in,mol}$ and $F_{out,mol}$ (mol s$^{-1}$) refer to the incoming and outgoing molar flow rates of dry air ("a" in subscript) or water vapour ("v" in subscript), $c_{pv}$ (J kg$^{-1}$ K$^{-1}$) is the constant-pressure heat capacity of water vapour, $T_{in}$ refers to the temperature of the incoming air (K) and $T_{out}$ refers to the temperature of the outgoing air (K). At steady state, $F_{out,mol,a} = F_{in,mol,a}$ and

$$F_{out,mol,w} = F_{in,mol,w} + L_A E_{lmol}, \tag{B2}$$

so we can solve the above for $H_l$ as:

$$H_l = E_{lmol} M_w T_{out} c_{pv}$$
$$+ \frac{(T_{out} - T_{in}) F_{in,mol,a} M_{air} c_{pa}}{L_A}$$
$$+ \frac{(T_{out} - T_{in}) F_{in,mol,w} M_w c_{pv}}{L_A} - \frac{Q_{in}}{L_A} \tag{B3}$$





The humidifier producing the air stream into the wind tunnel reported the volumetric flow rate of dry air at $T_r = 273.13$ K temperature and $P_r = 101300$ Pa pressure ($F_{in,v,a,n}$, m$^3$ s$^{-1}$) and the vapour pressure of the incoming air ($P_{w,in}$, Pa). To

convert from volumetric flow rates to molar flow rates, we used the ideal gas law ($P_a V_a = n_a R_{mol} T_a$), where the volume ($V_a$) was replaced by $F_{in,v,a,n}$, the molar amount ($n_a$) by $F_{in,mol,a}$ and $T_a$ and $P_a$ by their respective reference temperature and pressure ($T_r$ and $P_r$):

$$F_{in,mol,a} = \frac{P_r F_{in,v,a,n}}{R_{mol} T_r} \tag{B4}$$

The molar flow rate of water vapour is computed along similar lines:

$$F_{in,mol,w} = \frac{F_{in,v} P_{w,in}}{R_{mol} T_{in}} \tag{B5}$$

Considering Dalton's law of partial pressures, i.e. that the total pressure ($P_a$) is the sum of the partial pressures of water vapour and dry air, and assuming that both the dry air and the water vapour are well mixed within the same volume (represented by the volumetric flow rate into the chamber, $F_{in,v}$), we write

$$F_{in,v} = \frac{F_{in,mol,a} R_{mol} T_{in}}{P_a - P_{w,in}} \tag{B6}$$

To obtain the volumetric inflow rate ($F_{in,v}$) from the measured $F_{in,v,a,n}$, $P_{w,in}$ and $T_{in}$, we can substitute Eq. B4 into B6:

$$F_{in,v} = \frac{F_{in,v,a,n} P_r T_{in}}{(P_a - P_{w,in}) T_r} \tag{B7}$$

Given that $L_A$, $L_A E_{lmol}$, $T_{in}$, $T_{out}$, $Q_{in}$, $F_{in,v,a,n}$ and $P_{w,in}$ were measured directly, Eqs. B3–B6 could be used to infer the sensible heat flux ($H_l$) from the chamber energy balance, without any parameter fitting.

## B2 Inference of vapour pressure inside the wind tunnel

Similarly to Eq. B5, the molar outflow rate of water vapour was formulated as

$$F_{out,mol,w} = \frac{F_{out,v} P_{w,out}}{R_{mol} T_{out}} \tag{B8}$$

Equating Eq. B8 with B2, substitution of Eq. B5 and solving for $P_{w,out}$ gives the steady-state vapour pressure of the outgoing air as a function of the transpiration rate, the incoming air flow, its vapour pressure and temperature:

$$P_{w,out} = \frac{E_{l,mol} L_A R_{mol} T_{in} T_{out} + F_{in,v} P_{w,in} T_{out}}{F_{out,v} T_{in}} \tag{B9}$$

Due to changes in air temperature and humidity inside the wind tunnel, the volumetric outflow rate ($F_{out,v}$) does not necessarily equal the volumetric inflow rate ($F_{in,v}$). To calculate $F_{out,v}$, we used again the ideal gas law to obtain:

$$F_{out,v} = \frac{(F_{out,mol,a} + F_{out,mol,w}) R_{mol} T_{out}}{P_a} \tag{B10}$$





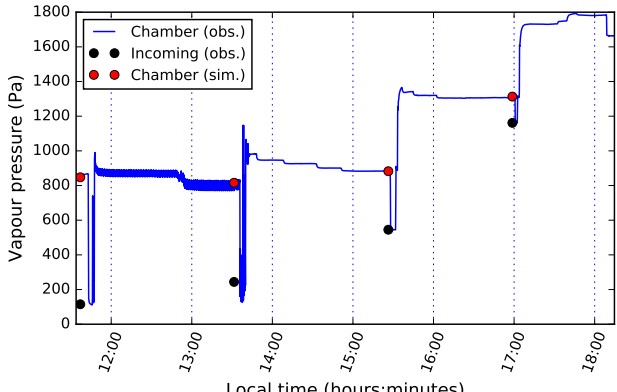

**Figure A5.** Time series of vapour pressure in the incoming and outgoing air. Experiment was conducted using an artificial leaf with a wet surface on the lower side, air temperature between 295.1 and 295.4 K and 3.9 m s$^{-1}$ wind speed. Outgoing air was passed through the LI-6400 XT IRGA and its vapour pressure was recorded continuously (blue line). At steady state, air flow was switched for a few minutes, so that the incoming air was passed through the IRGA (expressed as downwards steps in the blue line and marked by blue circles. The blue circles represent the average vapour pressures recorded for the incoming air, red circles represent the computed steady-state vapour pressure inside the chamber, using the chamber mass balance described in the main paper. The apparently thick line between 12:00 and 13:30 local time was the result of oscillations in vapour pressure caused by the control loop in the humidifier.

Considering that, at steady state, $F_{out,mol,a} = F_{in,mol,a}$, we substituted Eqs. B2, B4 and B5 into Eq. B10 to obtain $F_{out,v}$ as a function of directly measured quantities only:

$$F_{out,v} = \frac{1}{P_a T_r} \big( (E_{l,mol} L_A + F_{in,mol,w}) R_{mol} T_{out} T_r$$

$$+ F_{in,v,a,n} P_r T_{out} \big) \tag{B11}$$

Assuming well mixed air inside the wind tunnel, we assumed that $P_{wa} = P_{w,out}$ in our simulations and used Eqs. B9, B7 and B11 to calculate it. On some occasions, we attached an infrared gas analyser (LI6400XT, LI-COR, Lincoln, Nebraska, USA) to the outlet of the wind tunnel and verified results obtained from Eq. B9, without detecting any significant discrepancy (Fig. A5).

## B3 Measurement of net radiation of artificial leaf

We used net radiation sensors above and below the artificial leaf at a distance ($L_{ls}$) of 1 cm from the artificial leaf. Depending on the distance between the leaf and the sensors, as well as their sizes, the sensors only absorb a fraction of the radiation emitted by the leaf. For two rectangular plates, parallel and centred, this fraction ("view factor") can be calculated as (Incropera et al.,



2006, Table 13.1):

$$F_s = \frac{L_{ls}\left(\sqrt{\left(\frac{L_l}{L_{ls}} + \frac{L_s}{L_{ls}}\right)^2 + 4} - \sqrt{\left(\frac{L_l}{L_{ls}} - \frac{L_s}{L_{ls}}\right)^2 + 4}\right)}{2\,L_s} \qquad (B12)$$

where $L_l$ and $L_s$ are the widths of the leaf and the net radiation sensors, respectively. For 1 cm wide sensors 1 cm away from a 3 cm wide leaf, this fraction amounts to 0.82.

## Appendix C: Instruments and sensors

We used a range of commercial instruments and sensors in this study, some of which were modified and calibrated to fit our
purpose. The use of these sensors is described in the main text, with reference to their type. In Table A2, we provide further
details on the manufacturers and sensor specifications.



**Table A2.** Sensors and instruments used in this study.



| Function | Type and Manufacturer | Specifications |
|---|---|---|
| Liquid flow | SLI-0430, Sensirion AG, Staefa, Switzerland. http://www.sensirion.com | Max. flow rate: 50 $\mu$l min$^{-1}$ <br> Lowest calib. flow (LCF): 2 $\mu$l min$^{-1}$ <br> Accuracy above LCF: 5.0% of m.v. <br> Accuracy below LCF 0.2% of f.s. |
| Wind speed (calibration) | EE75, E + E Elektronik GmbH, Engerwitzdorf, Germany http://www.epluse.com | Range: 0.15–10 m s$^{-1}$ <br> Accuracy in air: $\pm 0.10$ m s$^{-1}$ + 1% of m.v. |
| Wind speed (monitoring) | FS5 Flowmodule, attached to IST evaluation board (P/N: 160.00001), Innovative Sensor Technology IST AG, Ebnat-Kappel, Switzerland http://www.ist-ag.com, | Range: 0–100 m s$^{-1}$ <br> Accuracy in air: $< 3\%$ of m.v. <br> Temperature sensitivity: $< 0.1\%$ K$^{-1}$ |
| Fan power supply | Programmable power supply 1786B, B&K Precision Corporation, Yorba Linda, CA 92887-4610, USA http://www.bkprecision.com | Resolution: 10 mV <br> Accuracy: $< 0.05\%$ + 10 mV |
| Fan power control | Bus-Powered Multifunction DAQ for USB, NI USB-6008 National Instruments Corporation, Austin, TX 78759-3504, USA, http://sine.ni.com | Inputs: 8 analog at 12 bits, up to 48 kS/s <br> Outputs: 2 analog, at 12 bits, software-timed. |
| Fan | MULTICOMP - MC35357 Farnell AG, 6300 Zug, Switzerland http://ch.farnell.com | Power: 12 V (DC) <br> Dimensions: $50 \times 50 \times 15$ mm <br> Max. flow rate: 0.526 m$^3$ min$^{-1}$ |
| Net radiation (calibration) | Net radiometer NR Lite2, Kipp & Zonen, Delft, The Netherlands http://www.kippzonen.com | Spectral range: 200 to 100.000 nm <br> Sensitivity: 10 $\mu$V per W m$^{-2}$ energy flux <br> Response time: $< 20$ s |
| Net radiation (monitoring) | gSKIN heat flux sensor, greenTEG, Zurich, Switzerland http://www.greenteg.com | Sensitivity: 1.9 $\mu$V per W m$^{-2}$ heat flux <br> Relative Error: $\pm 5\%$ <br> Sensor range: -10 to +10 kW m$^{-2}$ <br> Resolution: 0.5 W m$^{-2}$ |





| | Continued from previous page. | |
| --- | --- | --- |
| **Function** | **Type and Manufacturer** | **Specifications** |
| Temperature | T type thermocouples TG-T-30-SLE, TG-T-40-SLE, Omega Engineering GmbH 75392 Deckenpfronn, Germany http://www.omega.de | Temperature range: 0–350$^o$C Accuracy: ±0.5 K or 0.4% Wire diameter: 0.25 mm (30); 0.08 mm (40) |
| Temperature imaging | FLIR SC6000 FLIR Systems®, Inc. Wilsonville, OR, USA http://www.flir.com | Image resolution: 640×512 pixel Detector Type: QWIP Spectral range: 8–9.2 $\mu$m Sensitivity (NEDT): < 5 mK |
| Controlled air supply | Humidifier P-10C-0A-1-0-000100-v7 Cellkraft AB SE-114 19 Stockholm, Sweden http://www.cellkraft.se | Air flow range: 0–10 l min$^{-1}$ Flow accuracy: ±1% of f.s. Humidity range: 0–90% RH Humidity accuracy: ±1.7% of m.v. Temperature range: 4–300$^o$C Air supply: pressurised at 0–20 bar |
| Data logging | Data logger CR 1000 and 25-channel solid state multiplexer AM25T Campbell Scientific, Inc. Logan, UT 84321-1784, USA https://www.campbellsci.com | Analog Voltage Accuracy: ±0.06% of reading + offset Analog Resolution: 0.33 $\mu$V AD converter resolution: 13 bits |


m.v.: measured value
f.s.: full scale

## Appendix D: Additional experimental data

Figs. A6 and A7 illustrate additional experimental results obtained in the absence of shortwave irradiance for different perforation densities and varying wind speed or vapour pressure. Experimental conditions are summarised in the figure captions.




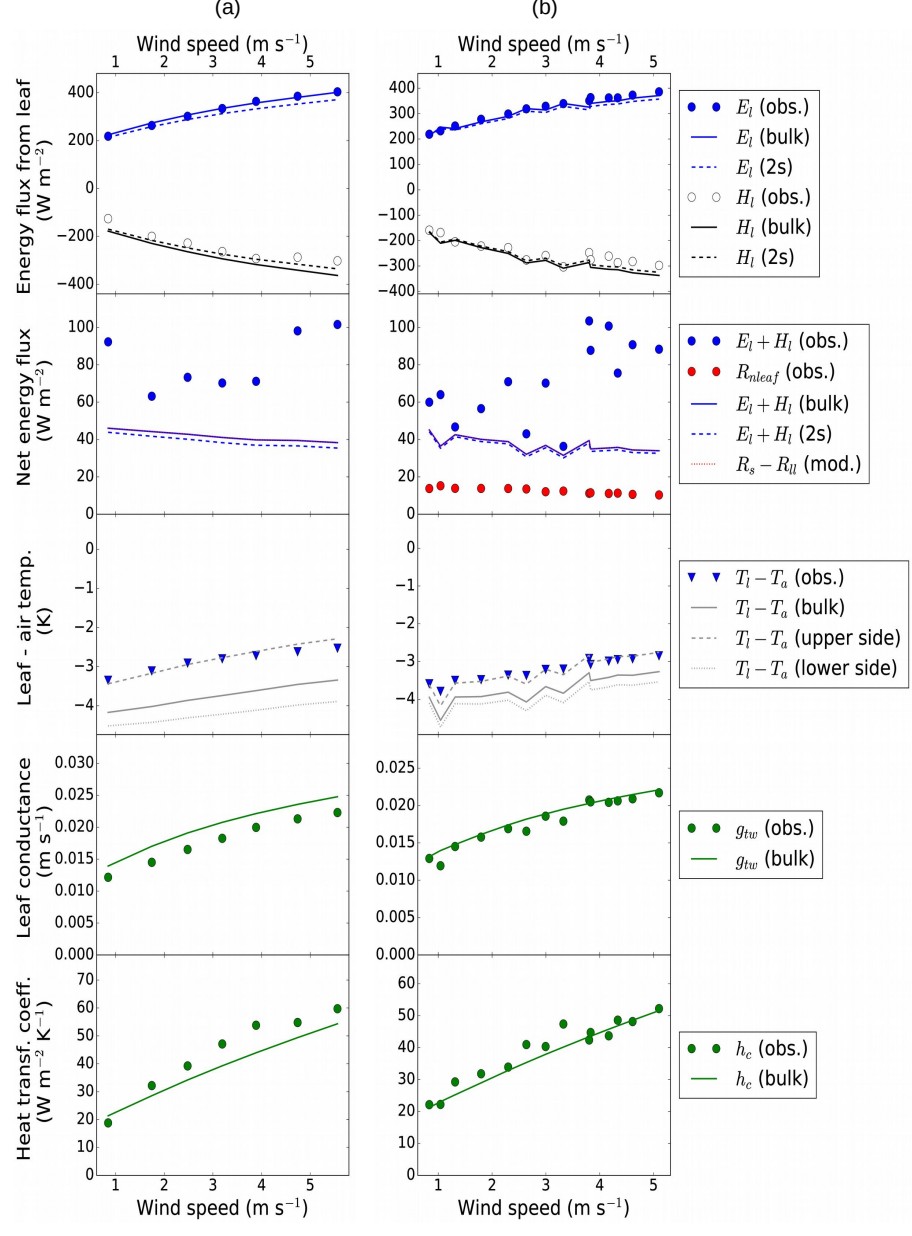

**Figure A6.** Artificial leaves with (a) 65 and (b) 35 pores mm$^{-2}$ under varying wind speed. Numerical model results (lines) based on the same boundary conditions as observations (symbols). $E_l$: latent heat flux; $H_l$: sensible heat flux; $R_s - R_{ll}$: absorbed net radiation; $T_l - T_a$: leaf-air temperature difference; $g_{tw}$: total leaf conductance to water vapour; $h_c$: convective heat transfer coefficient; "bulk": bulk leaf temperature model; "2s": model based on different leaf temperatures on both leaf sides. Boundary conditions: $g_{sw} = 0.05$ m s$^{-1}$ (a) and 0.042 m s$^{-1}$ (b); $R_s = 0$; $T_a = 295.1 - 295.3$ K (a) and 295.0–296.5 (b); $P_{wa} = 1200 - 1340$ Pa (a) and 1190–1280 (b); $k_l = 0.03$ W K$^{-1}$ m$^{-1}$ (a) and 0.1 W K$^{-1}$ m$^{-1}$ (b); $z_l = 0.35$ mm (a) and 0.6 mm (b).




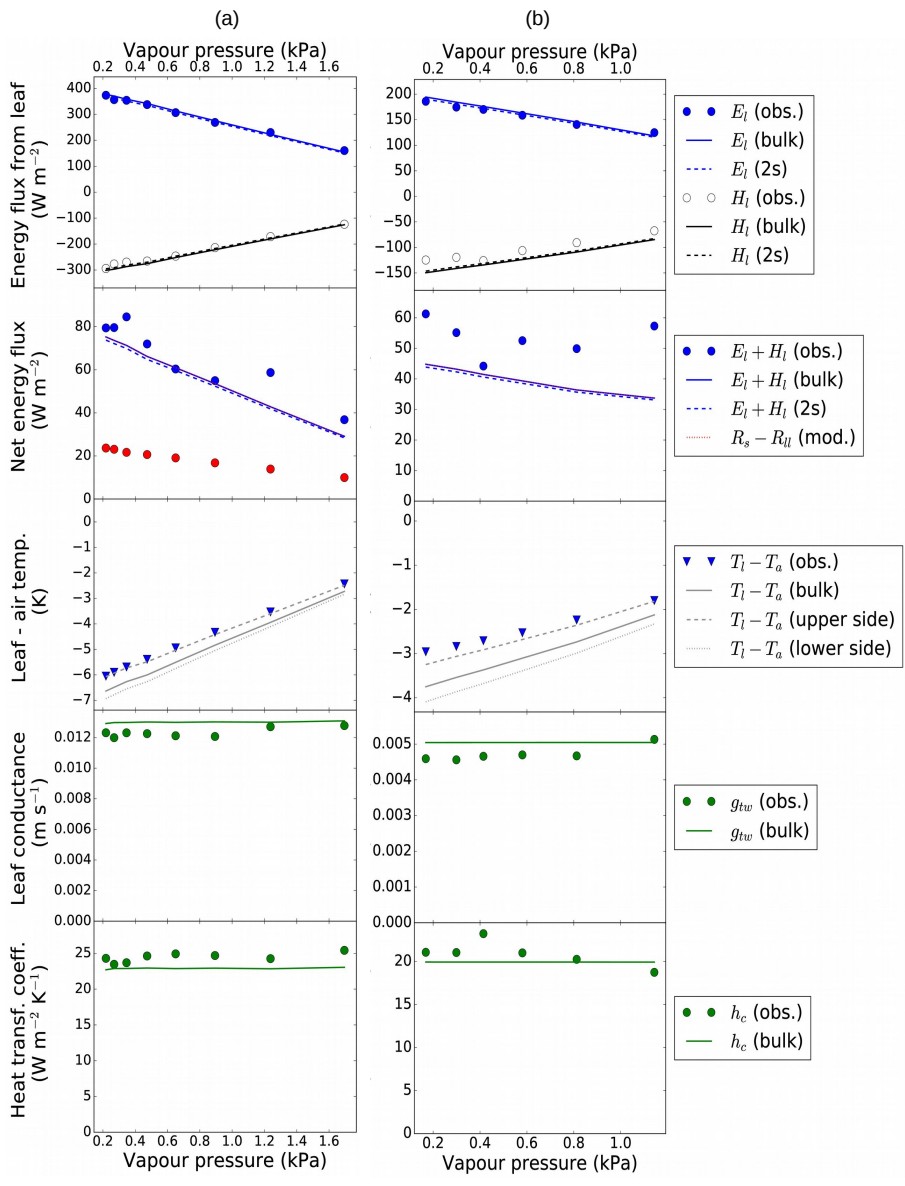

**Figure A7.** Artificial leaves with (a) 37 and (b) 7 pores mm$^{-2}$ under varying vapour pressure. Numerical model results (lines) based on the same boundary conditions as observations (symbols). $E_l$: latent heat flux; $H_l$: sensible heat flux; $R_s - R_{ll}$: absorbed net radiation; $T_l - T_a$: leaf-air temperature difference; $g_{tw}$: total leaf conductance to water vapour; $h_c$: convective heat transfer coefficient; "bulk": bulk leaf temperature model; "2s": model based on different leaf temperatures on both leaf sides. Boundary conditions: $g_{sw} = 0.035$ m s$^{-1}$ (a) and 0.007 m s$^{-1}$ (b); $R_s = 0$; $T_a = 295.7$–$296.0$ K (a) and $296.1$–$296.7$ K (b); $v_w = 1$ m s$^{-1}$ (a) and 0.7 m s$^{-1}$ (b); $k_l =0.1$ W K$^{-1}$ m$^{-1}$ (a) and 0.05 W K$^{-1}$ m$^{-1}$ (b); $z_l = 0.5$ mm.





**Table A3.** Table of symbols and standard values used in this paper. All area-related variables are expressed per unit leaf area.

| Variable | Description (value) | Units |
|---|---|---|
| $A_i$ | Conducting area of insulation material | $m^2$ |
| $A_p$ | Cross-sectional pore area | $m^2$ |
| $\alpha_l$ | Leaf albedo, fraction of shortwave radiation reflected by the leaf | 1 |
| $c_{pa}$ | Specific heat of dry air (1010) | $J\ K^{-1}\ kg^{-1}$ |
| $c_{pv}$ | Specific heat of water vapour at 300 K | $J\ K^{-1}\ kg^{-1}$ |
| $C_{wa}$ | Concentration of water in the free air | $mol\ m^{-3}$ |
| $C_{wl}$ | Concentration of water in the leaf air space | $mol\ m^{-3}$ |
| $d_p$ | Pore depth | m |
| $D_{va}$ | Binary diffusion coefficient of water vapour in air | $m^2\ s^{-1}$ |
| $E_l$ | Latent heat flux from leaf | $J\ m^{-2}\ s^{-1}$ |
| $E_{ll}$ | Latent heat flux from lower side of leaf | $J\ m^{-2}\ s^{-1}$ |
| $E_{l,mol}$ | Transpiration rate in molar units | $mol\ m^{-2}\ s^{-1}$ |
| $E_{lu}$ | Latent heat flux from upper side of leaf | $J\ m^{-2}\ s^{-1}$ |
| $\epsilon_l$ | Longwave emmissivity of the leaf surface (1.0) | 1 |
| $F_{in,mol,a}$ | Molar flow rate of dry air into chamber | $mol\ s^{-1}$ |
| $F_{in,mol,w}$ | Molar flow rate of water vapour into chamber | $mol\ s^{-1}$ |
| $F_{in,v}$ | Volumetric flow rate into chamber | $m^3\ s^{-1}$ |
| $F_{in,v,a,n}$ | Volumetric inflow of dry air at 0oC and 101325 Pa | $m^3\ s^{-1}$ |
| $F_{out,mol,a}$ | Molar flow rate of dry air out of chamber | $mol\ s^{-1}$ |
| $F_{out,mol,w}$ | Molar flow rate of water vapour out of chamber | $mol\ s^{-1}$ |
| $F_{out,v}$ | Volumetric flow rate out of chamber | $m^3\ s^{-1}$ |
| $F_s$ | Fraction of radiation emitted by leaf, absorbed by sensor | 1 |
| $g$ | Gravitational acceleration (9.81) | $m\ s^{-2}$ |
| $g_{bw}$ | Boundary layer conductance to water vapour | $m\ s^{-1}$ |
| $g_{bw,mol}$ | Boundary layer conductance to water vapour | $mol\ m^{-2}\ s^{-1}$ |
| $g_{sw}$ | Stomatal conductance to water vapour | $m\ s^{-1}$ |
| $g_{tw}$ | Total leaf conductance to water vapour | $m\ s^{-1}$ |
| $g_{tw,mol}$ | Total leaf layer conductance to water vapour | $mol\ m^{-2}\ s^{-1}$ |
| $h_c$ | Average 1-sided convective transfer coefficient | $J\ K^{-1}\ m^{-2}\ s^{-1}$ |
| $H_l$ | Sensible heat flux from leaf | $J\ m^{-2}\ s^{-1}$ |
| $H_{l_l}$ | Sensible heat flux from lower side of leaf | $J\ m^{-2}\ s^{-1}$ |






| Variable | Description (value) | Units |
|---|---|---|
| $H_{l_u}$ | Sensible heat flux from upper side of leaf | $\mathrm{J\ m^{-2}\ s^{-1}}$ |
| $k_a$ | Thermal conductivity of dry air | $\mathrm{J\ K^{-1}\ m^{-1}\ s^{-1}}$ |
| $k_{dv}$ | Ratio $D_{va}/V_m$ | $\mathrm{mol\ m^{-1}\ s^{-1}}$ |
| $k_l$ | Thermal conductivity of a fresh leaf | $\mathrm{J\ K^{-1}\ m^{-1}\ s^{-1}}$ |
| $L_A$ | Leaf area | $\mathrm{m^2}$ |
| $L_l$ | Characteristic length scale for convection (size of leaf) | m |
| $L_{ls}$ | Distance between leaf and net radiation sensor | m |
| $L_s$ | Width of net radiation sensor | m |
| $\lambda_E$ | Latent heat of evaporation (2.45e6) | $\mathrm{J\ kg^{-1}}$ |
| $M_{air}$ | Molar mass of air (kg mol-1) | $\mathrm{kg\ mol^{-1}}$ |
| $M_w$ | Molar mass of water (0.018) | $\mathrm{kg\ mol^{-1}}$ |
| $N_{Gr_L}$ | Grashof number | 1 |
| $N_{Le}$ | Lewis number | 1 |
| $N_{Nu_L}$ | Nusselt number | 1 |
| $n_p$ | Pore density | $\mathrm{m^{-2}}$ |
| $N_{Re_L}$ | Reynolds number | 1 |
| $N_{Re_c}$ | Critical Reynolds number for the onset of turbulence | 1 |
| $N_{Sh_L}$ | Sherwood number | 1 |
| $P_a$ | Air pressure | Pa |
| $P_r$ | Reference pressure | Pa |
| $P_{w,in}$ | Vapour pressure of incoming air | Pa |
| $P_{w,out}$ | Vapour pressure of outgoing air | Pa |
| $P_{wa}$ | Vapour pressure in the atmosphere | Pa |
| $P_{wl}$ | Vapour pressure inside the leaf | Pa |
| $N_{Pr}$ | Prandtl number (0.71) | 1 |
| $Q_{in}$ | Internal heat sources, such as fan | $\mathrm{J\ s^{-1}}$ |
| $Q_l$ | Conductive heat flux from upper to lower side of leaf | $\mathrm{J\ m^{-2}\ s^{-1}}$ |
| $R_d$ | Downwelling global radiation | $\mathrm{J\ m^{-2}\ s^{-1}}$ |
| $R_{ll}$ | Longwave radiation away from leaf | $\mathrm{J\ m^{-2}\ s^{-1}}$ |
| $R_{ll_l}$ | Longwave heat flux from lower side of leaf | $\mathrm{J\ m^{-2}\ s^{-1}}$ |
| $R_{ll_u}$ | Longwave heat flux from upper side of leaf | $\mathrm{J\ m^{-2}\ s^{-1}}$ |
| $R_{lu}$ | Upwards emitted/reflected global radiation from leaf | $\mathrm{J\ m^{-2}\ s^{-1}}$ |
| $R_{mol}$ | Molar gas constant (8.314472) | $\mathrm{J\ K^{-1}\ mol^{-1}}$ |



| Variable | Description (value) | Units |
|---|---|---|
| $R_s$ | Solar shortwave flux | $J\,m^{-2}\,s^{-1}$ |
| $R_u$ | Upwelling global radiation | $J\,m^{-2}\,s^{-1}$ |
| $r_p$ | Pore radius (for ellipsoidal pores, half the pore width) | m |
| $r_{sp}$ | Diffusive resistance of a stomatal pore | $s\,m^2\,mol^{-1}$ |
| $r_{vs}$ | Diffusive resistance of a stomatal vapour shell | $s\,m^2\,mol^{-1}$ |
| $S_a$ | Radiation sensor above leaf reading | $J\,m^{-2}\,s^{-1}$ |
| $S_b$ | Radiation sensor below leaf reading | $J\,m^{-2}\,s^{-1}$ |
| $S_s$ | Radiation sensor beside leaf reading | $J\,m^{-2}\,s^{-1}$ |
| $s_p$ | Spacing between stomata | m |
| $\sigma$ | Stefan-Boltzmann constant (5.67e-8) | $J\,K^{-4}\,m^{-2}\,s^{-1}$ |
| $T_a$ | Air temperature | K |
| $T_{in}$ | Temperature of incoming air | K |
| $T_l$ | Leaf temperature | K |
| $T_{l_l}$ | Leaf surface temperature of lower side | K |
| $T_{l_u}$ | Leaf surface temperature of upper side | K |
| $T_{out}$ | Temperature of outgoing air (= chamber $T_a$) | K |
| $T_r$ | Reference temperature | K |
| $T_w$ | Radiative temperature of objects surrounding the leaf | K |
| $V_m$ | Molar volume of air | $m^3\,mol^{-1}$ |
| $v_w$ | Wind velocity | $m\,s^{-1}$ |
| $z_l$ | Leaf thickness (m) | m |

*Author contributions.* SJS performed the mathematical derivations, designed and carried out the experiments and wrote the paper. DB designed and constructed major parts of the wind tunnel and custom sensor setup and contributed visuals and feedback for the manuscript. DO
was involved in the design of the experimental setup, interpretation of the results and writing the paper.

*Acknowledgements.* The authors are very grateful to Hans Wunderli for assistance in the lab, Stefan Meier and Joni Dehaspe for assistance in constructing artificial leaves and Ralph Beglinger (Lasergraph AG, Würenlingen, Switzerland) for laser perforation services. We also wish to acknowledge funding by the Swiss National Science Foundation, Project 2000021 135077.



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
