# Peer review of "Technical note: An experimental setup to measure latent and sensible heat fluxes from (artificial) plant leaves"

_Hydrology and Earth System Sciences, 2016_

## Referee Comment (RC1) · HHG Savenije (Referee) · 14 Feb 2017

This is the first time that I have seen an experimental setup that aims at determining all the components of the energy and the water balance of a leaf in a closed system where all parameters can be manipulated. As the authors indicate, the setup is not yet perfect in closing the energy balance and in preventing radiation to be reflected, but it is a great start. Of course we look forward to experiments with living leafs.

The authors wet our appetite by mentioning that their experiment led to a revision of the Penman Monteith equation, or actually a flaw in the P-M equation and referred to the paper by Schymanski and Or (2016). I think that the paper would benefit from not just providing the reference to this paper, but also in a few sentences mention the crux

of this flaw. That would certainly make the paper more attractive.

Minor comments: l.202: bracket missing

Figure 9: If 9b is based on 1m/s and 9a on 1.2-1.34 kPa, than the points on the left and the right should correspond for these values. This does not appear to be the case for the Net energy flux and the Leaf-air temperature and the graphs below that. Did I miss something?

---

## Author Comment (AC1) · 31 Mar 2017

Dear Prof. Savenije,

Thank you for your supportive review and for your suggestions. We will modify the sentence on L395 in the following way:

"In addition to the discovery of surprisingly strong temperature gradients between the two sides of a hypostomatous leaf (this study), previous experiments using the same setup have already led to the discovery of inconsistencies in the widely used Penman-Monteith equation for transpiration, mainly resulting from the neglect of two-sided sensible heat exchange by planar leaves (Schymanski and Or, 2017)."

We will also add the missing bracket in L.202, thanks for pointing this out.

We have inspected Fig. 9 again, and found that Panels (a) and (b) are a bit tricky to compare due to different axes scaling. To make the comparison easier, we re-drew the figure with common axes and added dashed lines to the points of correspondence (see attached Fig. 1). The points of correspondence should be at 1 m/s in 9a, i.e. slightly to the right of the left-most data points, and at 1.2-1.34 kPa in 9b, i.e. not on the right but roughly in the middle of the panel. At these points, the simulated values in 9a and 9b correspond very closely, whereas the observed net energy fluxes in 9a (second panel from top) deviate from the simulated by 60 W/m2, resulting in a mismatch between 9a and 9b in the observed values. We will explain this more clearly in the revised document.

Best regards,

Stan Schymanski, Dani Breitenstein and Dani Or

**Full figure caption for attached figure:** Artificial leaf with wet surface on the lower side (no stomatal resistance), under (a) varying wind speed and (b) varying vapour pressure. Numerical model results (lines) based on the same boundary conditions as observations (symbols). Red dashed lines indicate conditions in the plots where the forcing was roughly equivalent between Panels (a) and (b). $E_l$: latent heat flux; $H_l$: sensible heat flux; $R_s - R_{ll}$: absorbed net radiation; $T_l - T_a$: leaf-air temperature difference; $g_{tw}$: total leaf conductance to water vapour; $h_c$: convective heat transfer coefficient; "bulk": bulk leaf temperature model; "2s": model based on different leaf temperatures on both leaf sides. Boundary conditions: $g_{sw} = 999$ m s$^{-1}$; $R_s = 0$; $T_a = 295.4 - 295.6$ K (a) and $295.4 - 296.6$ (b); $P_{wa} = 1200 - 1342$ Pa (a); $v_w = 1.0$ m s$^{-1}$ (b); $k_l = 0.1$ W K$^{-1}$ m$^{-1}$; $z_l = 0.5$ mm.

[Figure]

[Figure]

**Fig. 1.** Revised Fig. 9. (See main text for full figure caption.)

---

## Referee Comment (RC2) · Anonymous Referee #2 · 29 Apr 2017

General comments This manuscript presents an experimental system to test our understanding of latent and sensible heat exchanges by leaves. The apparatus includes an artificial leaf that is intensely monitored to assess energy and mass (water) fluxes, allowing to compare prediction of conductance parameterizations and energy and mass balance equations with data obtained from a highly controlled setup. I find this contribution useful to pinpoint limitations in understanding and technical constraints on water and energy flow measurements. The manuscript is well written and thoroughly discussed and I only have some minor points to raise.

Specific comments The schematic figures 4 and 5 could be better integrated with the equations presented in the text, possibly presenting equation numbers in the figures

themselves. In general it would be useful to refer more explicitly to equation numbers in the text and figures (see some suggestions in the technical corrections). It would be worth checking and commenting on the recent paper by Zwieniecki et al. "Stomatal design principles in synthetic and real leaves". J. Royal Soc. Interface, DOI: 10.1098/rsif.2016.0535.

Technical corrections L85: delete extra "in" L178: refer to the Appendix equations L187: refer to Equation 1 regarding the energy balance Eq. 9: delete extra brackets in the numerator; add closing bracket after units in L202 L444: move this definition of np up to eq. A1, where the symbol is first used Table A1 (caption): missing reference to a figure (appears as "??") Figures A2 and A3: the lines in the insets with black background are hard to see L502: "give" not "gives" (plural subject)

---

## Author Comment (AC2) · 4 May 2017

We would like to thank the reviewer for the very positive assessment of the manuscript and helpful suggestions for further improvements.

Concerning Fig. 4, we realised that we did not actually provide the equations to compute net radiation from the sensor signals, which will be done in the revised manuscript by extending Appendix B3. We will refer to the equations in the figure caption. Thank you for pointing us to this omission.

Fig. 5 displays simplified equations for illustration of the general principle, whereas Appendix B1 contains the detailed equations. We will refer to the detailed equations in

the figure caption, but we feel that it would overburden the figure and make the general approach less obvious if we provided the detailed equations in the figure.

We will go through the text again and include more explicit references to the respective equations where appropriate, also following the advice given in the detailed comments, which are very helpful. In this context, we would like to thank the referee for the very clear suggestions in the detailed comments, which will all be adopted in the revised manuscript. We will also cite Zwieniecki et al. (2016) along with the older citations for previous experiments with perforated surfaces, as this paper shows nicely that even today, experiments usually consider pores at scales orders of magnitude larger than stomata. We were not able to extract any additional relevant insights from the paper, as it does not discuss the leaf energy balance, which is the main focus of our manuscript.

Best regards,

Stan Schymanski, on behalf of all co-authors

---

## Author Response (AR1)

**Description of changes in revised manuscript**

Stan Schymanski

June 2, 2017

Dear editor, dear referees,

We would like to thank you again for all the helpful comments and for giving us the opportunity to improve our manuscript. We implemented all the changes we promised in our original responses to the reviewers and added a few additional improvements, as listed below. In the below summary of changes, we refer to the attached version of the manuscript with highlighted changes (note the different line numbers compared to the revised manuscript itself).

We hope that the revised manuscript satisfies the high quality standards of HESS and look forward to your response.

- L36–37: Removed "micro-" and added reference to Zwieniecki (2016), to reflect that previous experiments often included perforations that were orders of magnitude larger than stomata. (Referee 2)

- L85: Corrected language.

- L114: added reference to Appendix for detailed computations. (Referee 2)

- L139: Moved figure to appendix and added equations explaining the terms in the figure there. (Referee 2)

- Fig. 4: Added reference to equations in appendix. (Referee 2)

- L172: Added reference to equations.

- L176: Fixed reference.

- L182: Added reference to equations.

- L188: Fixed reference.

- L190: Added reference to equation.

- L195: Fixed reference.

- Eq. 9: Removed brackets (Referee 2)

- L207: Added bracket (Referee 2)

- L214: Added reference to equation.

- Fig. 8: Added lines of correspondence between figures to improve clarity. (Referee 1)

- Figs. 8, 9, A7 and A8: Modified legend and caption for more clarity.

- L400: Modified text about previous findings. (Referee 1)

- L442: Moved definitions of variables to where they are first mentioned. (Referee 2)

- L446: Fixed notation.

- L449: Fixed reference.

- Table A1: Fixed reference to figures in caption. (Referee 2)

- Fig. A2: Changed colour and thickness of lines. (Referee 2)

- L510: Fixed language. (Referee 2)

- L525–546: Added description and equations related to leaf radiative balance. (Referee 2)

[revised manuscript text omitted]